# $f$-Policy Gradients: A General Framework for Goal Conditioned RL using $f$-Divergences

**Siddhant Agarwal**
The University of Texas at Austin
siddhant@cs.utexas.edu

**Ishan Durugkar**
Sony AI
ishan.durugkar@sony.com

**Peter Stone**
The University of Texas at Austin
Sony AI
pstone@cs.utexas.edu

**Amy Zhang**
The University of Texas at Austin
amy.zhang@austin.utexas.edu

## Abstract

Goal-Conditioned Reinforcement Learning (RL) problems often have access to sparse rewards where the agent receives a reward signal only when it has achieved the goal, making policy optimization a difficult problem. Several works augment this sparse reward with a learned dense reward function, but this can lead to sub-optimal policies if the reward is misaligned. Moreover, recent works have demonstrated that effective shaping rewards for a particular problem can depend on the underlying learning algorithm. This paper introduces a novel way to encourage exploration called $f$-Policy Gradients, or $f$-PG. $f$-PG minimizes the f-divergence between the agent's state visitation distribution and the goal, which we show can lead to an optimal policy. We derive gradients for various f-divergences to optimize this objective. Our learning paradigm provides dense learning signals for exploration in sparse reward settings. We further introduce an entropy-regularized policy optimization objective, that we call $state$-MaxEnt RL (or $s$-MaxEnt RL) as a special case of our objective. We show that several metric-based shaping rewards like L2 can be used with $s$-MaxEnt RL, providing a common ground to study such metric-based shaping rewards with efficient exploration. We find that $f$-PG has better performance compared to standard policy gradient methods on a challenging gridworld as well as the Point Maze and FetchReach environments. More information on our website https://agarwalsiddhant10.github.io/projects/fpg.html.

## 1 Introduction

Reinforcement Learning (RL) algorithms aim to identify the optimal behavior (policy) for solving a task by interacting with the environment. The field of RL has made large strides in recent years (Mnih et al., 2013; Silver et al., 2017; Haarnoja et al., 2018; Ouyang et al., 2022; Wurman et al., 2022) and has been applied to complex tasks ranging from robotics (Gupta et al., 2019), protein synthesis (Jumper et al., 2021), computer architecture (Fawzi et al., 2022) and finance (Liu et al., 2021). Goal-Conditioned RL (GCRL) is a generalized form of the standard RL paradigm for learning a policy that can solve many tasks, as long as each task can be defined by a single rewarding goal state. Common examples of goal-conditioned tasks arise in robotics where the goal states can be a target object configuration for manipulation-based tasks (Kim et al., 2022; Gupta et al., 2019; OpenAI et al., 2021) or a target location for navigation-based tasks (Shah et al., 2020; Gervet et al., 2023).

In any reinforcement learning setup, the task is conveyed to the agent using rewards (Silver et al., 2021). In goal-conditioned RL settings, a common reward function used is 1 when the goal is

37th Conference on Neural Information Processing Systems (NeurIPS 2023).

achieved and 0 everywhere else. This reward function is sparse and poses a huge learning challenge to obtain the optimal policy without any intermediate learning signal. Prior works (Ng et al., 1999; Ni et al., 2020; Durugkar et al., 2021; Arjona-Medina et al., 2019; Goyal et al., 2019) have augmented the reward function to provide some dense signal for policy optimization. A major issue with augmenting reward functions is that the optimal policy for the new reward function may no longer be optimal under the original, true reward function (Ng et al., 1999). Moreover, it has been shown (Booth et al., 2023) that shaping rewards that improve learning for one learning algorithm may not be optimal for another learning algorithm. Algorithms that learn reward functions (Ni et al., 2020; Durugkar et al., 2021; Zheng et al., 2018) are inefficient because the reward function must first be learned before it can be used for policy optimization. These challenges lead to the following research question: *Is there another way to provide dense learning signals for policy optimization other than through dense shaping rewards?*

In this work, we look at using divergence minimization between the agent's state visitation and the goal distribution (we assume that each goal can be represented as a distribution, Dirac distribution being the simplest) as an objective to provide additional learning signals. Similar perspectives to policy learning has been explored by prior works (Ziebart et al., 2008; Haarnoja et al., 2017, 2018; Ho & Ermon, 2016; Ni et al., 2020; Ghasemipour et al., 2019; Fu et al., 2017), but they reduce their methods into a reward-centric view. MaxEnt RL methods (Ziebart et al., 2008; Haarnoja et al., 2017, 2018) use the distribution over trajectories rather than state visitations and still suffer from sparsity if the task rewards are sparse. Imitation learning works like those of Ho & Ermon (2016); Fu et al. (2017); Ghasemipour et al. (2019) use a variational lower bound to obtain min-max objectives that require discriminators. These objectives suffer from mathematical instabilities and often require coverage assumptions i.e., abundant overlap between the agent's state visitation distribution and goal distribution. Our method does not rely on discriminators nor does it assume state coverage. It provides dense signals to update the policy even when the agent has not seen the goal. These signals push the policy towards higher entropy state visitations until the goal is discovered.

Our method, $f$-**PG or** $f$-**Policy Gradient**, introduces a novel GCRL framework that aims to minimize a general measure of mismatch (the $f$-divergence) between the agent's state visitation distribution and the goal distribution. We prove that minimizing the $f$-divergence (for some divergences) recovers the optimal policy. The analytical gradient for the objective looks very similar to a policy gradient which allows us to leverage established methods from the policy gradient literature to come up with an efficient algorithm for goal-conditioned RL. We show the connection of our method to the commonly used metric-based shaping rewards for GCRL like L2 rewards. We show that a special case of $f$-PG jointly optimizes for maximization of a reward and the entropy of the state-visitation distribution thus introducing **state-MaxEnt RL** (or **s-MaxEnt RL**). Using a sparse gridworld, we establish the benefits of using $f$-PG as a dense signal to explore when the agent has not seen the goal. We also demonstrate that our framework can be extended to continuous state spaces and scale to larger and higher-dimensional state spaces in maze navigation and manipulation tasks.

Our key contributions are 1) developing a novel algorithm for goal-conditioned RL that provably produces the optimal policy, 2) connecting our framework to commonly known metric-based shaping rewards, 3) providing a new perspective to RL ($s$-MaxEnt RL) that focuses on maximizing the entropy of the state-visitation distribution and 4) empirical evidence demonstrating its ability to provide dense learning signals and scale to larger domains.

## 2 Background

This section goes over the standard goal-conditioned reinforcement learning formulation and the f-divergences that will be used in the rest of the paper.

**Goal-conditioned reinforcement learning.** This paper considers an agent in a goal-conditioned MDP (Puterman, 1990; Kaelbling, 1993). A goal-conditioned MDP is defined as a tuple $\langle \mathcal{S}, \mathcal{G}, \mathcal{A}, P, r, \gamma, \mu_0, \rho_g \rangle$ where $\mathcal{S}$ is the state space, $\mathcal{A}$ is the action space, $P : \mathcal{S} \times \mathcal{A} \longmapsto \Delta(\mathcal{S})$ is the transition probability ($\Delta(\cdot)$ denotes a probability distribution over a set), $\gamma \in [0, 1)$ is the discount factor, $\mu_0$ is the distribution over initial states, $\mathcal{G} \subset \mathcal{S}$ is the set of goals, and $\rho_g : \Delta(\mathcal{G})$ is the distribution over goals. At the beginning of an episode, the initial state $s_0$ and the goal $g$ are sampled from the distributions $\mu_0$ and $\rho_g$. The rewards $r : \mathcal{S} \times \mathcal{G} \longmapsto \mathbb{R}$ are based on the state the agent visits and conditioned on the goal specified during that episode. This work focuses on sparse rewards, where $r(s', g) = 1$ when $s' = g$, and is $r(s', g) = 0$ otherwise. In continuous domains, the equality is relaxed to $s' \in \mathcal{B}(g, r)$ where $\mathcal{B}(g, r)$ represents a ball around the goal $g$ with radius $r$.

A trajectory $\tau$ is defined as the sequence $(s_0, a_0, s_1, \ldots, s_{T-1}, a_{T-1}, s_T)$. The return $H_g(s)$ is defined as the cumulative undiscounted rewards $H_g(s) := \sum_{t=0}^{T} [r(s_{t+1}, g)|s_0 = s]$, where $T$ is the length of a trajectory. We will assume the trajectory ends when a maximum number of policy steps $(T)$ have been executed. The agent aims to learn a policy $\pi : \mathcal{S} \times \mathcal{G} \longmapsto \Delta(\mathcal{A})$ that maximises the expected return $\mathbb{E}_{\pi, s_0}[H_g(s_0)]$. The optimal policy $\pi^* = \arg\max_{\pi_\theta \in \Pi} \mathbb{E}_{\pi, s_0}[H_g(s_0)]$, where the space of policies $\Pi$ is defined by a set of parameters $\theta \in \Theta$.

**Distribution matching approach to goal-conditioned RL.** The distribution over goal-conditioned trajectories is defined as $p_\theta(\tau; g) = p(s_0)\Pi_{t=0}^{T}p(s_t|s_{t-1}, a_{t-1})\pi_\theta(a_t|s_t; g)$. The trajectory-dependent state visitation distribution is defined as $\eta_\tau(s)$. It is the number of times the state $s$ is visited in the trajectory $\tau$. The agent's goal-conditioned state visitation can then be defined as:

$$p_\theta(s; g) = \frac{\int p_\theta(\tau; g)\eta_\tau(s)d\tau}{Z} \tag{1}$$

$$= \frac{\int \Pi p(s_{t+1}|s_t, a_t)\pi_\theta(a_t|s_t; g)\eta_\tau(s)}{\int \int \Pi p(s_{t+1}|s_t, a_t)\pi_\theta(a_t|s_t; g)\eta_\tau(s)d\tau ds}d\tau. \tag{2}$$

The goal $g$ defines an idealized target distribution $p_g : \Delta(\mathcal{S})$, considered here as a Dirac distribution which places all the probability mass at the goal state $p_g = \delta(g)$. Such a formulation has been used previously in approaches to learn goal-conditioned policies (Durugkar et al., 2021). This work focuses on minimizing the mismatch of an agent's goal-conditioned state visitation distribution $p_\theta(s; g)$ to this target distribution $p_g$. In this paper, we will be using $p_\theta$ and $p_\pi$ interchangeably i.e., $p_\theta$ corresponds to the visitation distribution induced by policy $\pi$ that is parameterized by $\theta$.

To do so, this paper considers a family of methods that compare the state-visitation distribution induced by a goal-conditioned policy and the ideal target distribution for that goal $g$, called $f$-divergences. $f$-divergences are defined as (Polyanskiy & Wu, 2022),

$$D_f(P||Q) = \int_{P>0} P(x)f\Big(\frac{Q(x)}{P(x)}\Big)dx - f'(\infty)Q([P(x) = 0]), \tag{3}$$

where $f$ is a convex function with $f(1) = 0$. $f'(\infty)$ is not defined (is $\infty$) for several $f$-divergences and so it is a common assumption that $Q = 0$ wherever $P = 0$. Table 1 shows a list of commonly used $f$-divergences with corresponding $f$ and $f'(\infty)$.

| $f$-divergence | $D_f(P||Q)$ | $f(u)$ | $f'(u)$ | $f'(\infty)$ |
|:---:|:---:|:---:|:---:|:---:|
| **FKL** | $\int P(x)\log\frac{P(x)}{Q(x)}dx$ | $u\log u$ | $1 + \log u$ | Undefined |
| **RKL** | $\int Q(x)\log\frac{Q(x)}{P(x)}dx$ | $-\log u$ | $-\frac{1}{u}$ | 0 |
| **JS** | $\frac{1}{2}\int P(x)\log\frac{2P(x)}{P(x)+Q(x)} +$ $Q(x)\log\frac{2Q(x)}{P(x)+Q(x)}dx$ | $u\log u-$ $(1 + u)\log\frac{1+u}{2}$ | $\log\frac{2u}{1+u}$ | $\log 2$ |
| $\chi^2$ | $\frac{1}{2}\int Q(x)(\frac{P(x)}{Q(x)} - 1)^2 dx$ | $\frac{1}{2}(u - 1)^2$ | $u$ | Undefined |

Table 1: Selected list of $f$-divergences $D_f(P||Q)$ with generator functions $f$ and their derivatives $f'$, where $f$ is convex, lower-semicontinuous and $f(1) = 0$.

## 3 Related Work

**Shaping Rewards.** Our work is related to a separate class of techniques that augment the sparse reward function with dense signals. Ng et al. (1999) proposes a way to augment reward functions without changing the optimal behavior. Intrinsic Motivation (Durugkar et al., 2021; Bellemare et al., 2016; Singh et al., 2010; Barto, 2013) has been an active research area for providing shaping rewards. Some work (Niekum, 2010; Zheng et al., 2018) learn intrinsic or alternate reward functions for the underlying task that aim to improve agent learning performance while others (Durugkar et al., 2021; Ni et al., 2020; Goyal et al., 2019) learn augmented rewards based on distribution matching. AIM (Durugkar et al., 2021) learns a potential-based shaping reward to capture the time-step distance but requires a restrictive assumption about state coverage, especially around the goal while we do not make any such assumption. Recursive classification methods (Eysenbach et al., 2021, 2020) use future state densities as rewards. However, these methods will fail when the agent has never seen the goal. Moreover, in most of these works, the reward is not stationary (is dependent on the policy)

which can lead to instabilities during policy optimization. GoFAR (Ma et al., 2022) is an offline goal-conditioned RL algorithm that minimizes a lower bound to the KL divergence between $p_\theta(s)$ and the $p_g(s)$. It computes rewards using a discriminator and uses the dual formulation utilized by the DICE family (Nachum et al., 2019), but reduces to GAIL (Ho & Ermon, 2016) in the online setting, requiring coverage assumptions. Our work also minimizes the divergence between the agent's visitation distribution and the goal distribution, but we provide a new formulation for on-policy goal-conditioned RL that does not require a discriminator or the same coverage assumptions.

**Policy Learning through State Matching.** We first focus on imitation learning where the expert distribution $p_E(s, a)$ is directly inferred from the expert data. GAIL (Ho & Ermon, 2016) showed that the inverse RL objective is the dual of state-matching. f-MAX (Ghasemipour et al., 2019) uses f-divergence as a metric to match the agent's state-action visitation distribution $p_\pi(s, a)$ and $p_E(s, a)$. Ke et al. (2019); Ghasemipour et al. (2019) shows how several commonly used imitation learning methods can be reduced to a divergence minimization. But all of these methods optimize a lower bound of the divergence which is essentially a min-max bilevel optimization objective. They break the min-max into two parts, fitting the density model to obtain a reward that can be used for policy optimization. But these rewards depend on the policy, and should not be used by RL algorithms that assume stationary rewards. f-IRL (Ni et al., 2020) escapes the min-max objective but learns a reward function that can be used for policy optimization. *We do not aim to learn a reward function but rather directly optimize for a policy using dense signals from an $f$-divergence objective.*

In reinforcement learning, the connections between entropy regularized MaxEnt RL and the minimization of reverse KL between agent's trajectory distribution, $p_\pi(\tau)$, and the "optimal" trajectory distribution, $p^*(\tau) \propto e^{r(\tau)}$ has been extensively studied Ziebart (2010); Ziebart et al. (2008); Kappen et al. (2012); Levine (2018); Haarnoja et al. (2018). MaxEnt RL optimizes for a policy with maximum entropy but such a policy does not guarantee maximum coverage of the state space. Hazan et al. (2018) discusses an objective for maximum exploration that focuses on maximizing the entropy of the state-visitation distribution or KL divergence between the state-visitation distribution and a uniform distribution. A few works like Durugkar et al. (2023, 2021); Ma et al. (2022), that have explored state-matching for reinforcement learning, have been discussed above. Several works like (Belousov & Peters, 2018, 2019; Touati et al., 2020) have used divergence to constraint the policy improvement steps making the updates more stable.

**Limitations of Markov Rewards.** Our work looks beyond the maximization of a Markov reward for policy optimization. The learning signals that we use are non-stationary. We thus discuss the limitations of using Markov rewards for obtaining the optimal policy. There have been works (Abel et al., 2021; Clark & Amodei, 2016; Icarte et al., 2018, 2021) that express the difficulty in using Markov rewards. Abel et al. (2021) proves that there always exist environment-task pairs that cannot be described using Markov rewards. Reward Machines (Icarte et al., 2018) create finite automata to specify reward functions and can specify Non-Markov rewards as well but these are hand-crafted.

## 4    $f$-Policy Gradient

In this paper, we derive an algorithm where the agents learn by minimizing the following $f$-divergence:

$$J(\theta) = D_f(p_\theta(s)||p_g(s)) \tag{4}$$

In this section, we shall derive an algorithm to minimize $J(\theta)$ and analyze the objective more closely in the subsequent section. Unlike f-max (Ghasemipour et al., 2019), we directly optimize $J(\theta)$. We differentiate $J(\theta)$ with respect to $\theta$ to get this gradient. We follow a similar technique as Ni et al. (2020) to obtain the analytical gradient of $J(\theta)$.

**Theorem 4.1.** *The gradient of $J(\theta)$ as defined in Equation 4 is given by,*

$$\nabla_\theta J(\theta) = \mathbb{E}_{\tau \sim p_\theta(\tau)} \left[ \left[ \sum_{t=1}^{T} \nabla_\theta \log \pi_\theta(a_t|s_t) \right] \left[ \sum_{t=1}^{T} f'\left(\frac{p_\theta(s_t)}{p_g(s_t)}\right) \right] \right]. \tag{5}$$

The gradient looks exactly like policy gradient with rewards $-f'\left(\frac{p_\theta(s_t)}{p_g(s_t)}\right)$. However, this does not mean that we are maximizing $J^{RL}(\theta) = \mathbb{E}_{\tau \sim p_\theta(\tau)} \left[ -f'\left(\frac{p_\theta(s_t)}{p_g(s_t)}\right) \right]$. This is because the gradient of $J^{RL}(\theta)$ is not the same as $\nabla_\theta J(\theta)$. For Dirac goal distributions, the gradient in Equation 5 cannot be

used (as $f'\left(\frac{p_\theta(s_t)}{p_g(s_t)}\right)$ will not be defined when $p_g(s_t) = 0$). We can use the definition of $f$-divergence in Equation 3 to derive a gradient for such distributions.

The gradient is obtained in terms of the state visitation frequencies $\eta_\tau(s)$. Further examination of the gradient leads to the following theorem,

**Theorem 4.2.** *Updating the policy using the gradient (Equation 5) maximizes* $\mathbb{E}_{p_\theta}[\eta_\tau(g)]$.

Theorem 4.2 provides another perspective for $f$-Policy Gradient – $\eta_\tau(g)$ is equivalent to the expected return for a goal-based sparse reward, hence optimizing the true goal-conditioned RL objective. We shall prove the optimality of the policy obtained from minimizing $J(\theta)$ in the next section.

In practice, a Dirac goal distribution can be approximated by clipping off the zero probabilities at $\epsilon$, similar to Laplace correction. Doing so, we will be able to use dense signals from the gradient in Equation 5 while still producing the optimal policy. This approximation is different from simply adding an $\epsilon$ reward at every state. This is because the gradients are still weighed by $f'\left(\frac{p_\theta(s_t)}{\epsilon}\right)$ which depends on $p_\theta(s_t)$.

Simply optimizing $J(\theta)$ is difficult because it faces similar issues to REINFORCE (Williams & Peng, 1991). A major shortcoming of the above gradient computation is that it requires completely on-policy updates. This requirement will make learning sample inefficient, especially when dealing with any complex environments. However, there have been a number of improvements to naïve policy gradients that can be used. One approach is to use importance sampling (Precup, 2000), allowing samples collected from a previous policy $\pi_{\theta'}$ to be used for learning. To reap the benefits of importance sampling, we need the previous state-visitation distributions to compute $f'\left(\frac{p_\theta(s)}{p_g(s)}\right)$. Hence, we need to ensure that the current policy does not diverge much from the previous policy. This condition is ensured by constraining the KL divergence between the current policy and the previous policy. We use the clipped objective similar to Proximal Policy Optimization (Schulman et al., 2017), which has been shown to work well with policy gradients. PPO has shown that the clipped loss works well even without an explicit KL constraint in the objective. The gradient used in practice is,

$$\nabla_\theta J(\theta) = \mathbb{E}_{s_t, a_t \sim p_{\theta'}(s_t, a_t)}\left[\min(r_\theta(s_t)F_{\theta'}(s_t), clip(r_\theta(s_t), 1 - \epsilon, 1 + \epsilon)F_{\theta'}(s_t))\right] \quad (6)$$

where $r_\theta(t) = \frac{\pi_\theta(a_t|s_t)}{\pi_{\theta'}(a_t|s_t)}$ and $F_{\theta'}(s_t) = \sum_{t'=t}^{T} \gamma^{t'} f'\left(\frac{p_{\theta'}(s_t)}{p_g(s_t)}\right)$. The derivation for this objective is provided in Appendix B. $\gamma$ is added to improve the stability of gradients and to prevent the sum of $f'\left(\frac{p_{\theta'}(s_t)}{p_g(s_t)}\right)$ from exploding.

For the purpose of this paper, we use kernel density estimators to estimate the goal distribution and the agent's state visitation distribution. We may also use discriminators to estimate the ratio of these densities like Ho & Ermon (2016); Fu et al. (2017); Ghasemipour et al. (2019). But unlike these methods, we will not be incorrectly breaking a minmax objective. In our case, the estimate of the gradient requires the value of the ratio of the two distributions and does not make any assumptions about the stationarity of these values. While the adversarial methods break the minmax objective and assume the discriminator

---

**Algorithm 1** $f$-PG

---

Let, $\pi_\theta$ be the policy, $G$ be the set of goals, $B$ be a buffer
**for** $i = 1$ **to** num_iter **do**
  $B \leftarrow []$
  **for** $j = 1$ **to** num_traj_per_iter **do**
    Sample $g$, set $p_g(s)$
    Collect goal conditioned trajectories, $\tau : g$
    Fit $p_\theta(s)$ using KDE on $\tau$
    Store $f'\left(\frac{p_\theta(s)}{p_g(s)}\right)$ for each $s$ in $\tau$
    $B \leftarrow B + \{\tau : g\}$
  **end for**
  **for** $j = 1$ **to** num_policy_updates **do**
    $\theta \leftarrow \theta - \alpha \nabla_\theta J(\theta)$ (Equation 6)
  **end for**
**end for**

---

to be fixed (and rewards stationary) during policy optimization.

## 5 Theoretical analysis of $f$-PG

In this section, we will first show that minimizing the f-divergence between the agent's state visitation distribution and goal distribution yields the optimal policy. We will further analyze the connections to metric based shaping rewards and implicit exploration boost from the learning signals. For the rest

of the paper, we will refer to $f$-PG using FKL divergence as $fkl$-PG, $f$-PG using RKL divergence as $rkl$-PG and so on.

## 5.1 Analysis of $J(\theta)$

This section shows that the policy obtained by minimizing an $f$-divergence between the agent's state visitation distribution and the goal distribution is the optimal policy.

**Theorem 5.1.** *The policy that minimizes $D_f(p_\pi || p_g)$ for a convex function $f$ with $f(1) = 0$ and $f'(\infty)$ being defined, is the optimal policy.*

The proof for Theorem 5.1 is provided in Appendix A. The Theorem states that the policy obtained by minimizing the $f$-divergence between the agent's state-visitation distribution and the goal distribution is the optimal policy for a class of convex functions defining the $f$-divergence with $f'(\infty)$ defined. It thus makes sense to minimize the $f$-divergence between the agent's visitation and the goal distribution. It must be noted that the objective does not involve maximizing a reward function. Note that the condition that $f'(\infty)$ is defined is not true for all $f$-divergences. The common $f$-divergences like RKL, TV, and JS have $f'(\infty)$ defined $rkl$-PG, $tv$-PG, and $js$-PG will produce the optimal policy.

Forward KL divergence (FKL) has $f = u \log u$ and so does not have $f'(\infty)$ defined. Does this mean that the policy obtained by minimizing the FKL divergence is not optimal? Lemma 5.1 (proof in Appendix A) shows that the policy obtained maximizes the entropy of the agent's state-visitation distribution along with maximizing a reward of $\log p_g(s)$.

**Lemma 5.1.** *$fkl$-PG produces a policy that maximizes the reward $\log p_g(s)$ along with the entropy of the state-visitation distribution.*

A similar result can be shown for $\chi^2$-divergence as well. It must be understood that Lemma 5.1 does not mean that $fkl$-PG is the same as the commonly studied MaxEnt RL.

**Differences from MaxEnt RL**: MaxEnt RL, as studied in Haarnoja et al. (2017, 2018), maximizes the entropy of the policy along with the task reward to achieve better exploration. However, maximizing the entropy of the policy does not imply maximum exploration. Hazan et al. (2018) shows that maximizing the entropy of the state-visitation distribution provably provides maximum exploration. Lemma 5.1 shows that $fkl$-PG maximizes the entropy of the state-visitation distribution along with the reward making it better suited for exploration. To distinguish our work, we call the MaxEnt RL, as discussed in works like Haarnoja et al. (2017, 2018), as $\pi$-**MaxEnt RL** because it only focuses on the entropy of the policy. On the other hand, $fkl$-PG maximizes the entropy of the state-visitation distribution so we call it **state-MaxEnt RL** or **s-MaxEnt RL**. Similarly, **sa-MaxEnt RL** can be defined to maximize the entropy of the state-action visitation distribution.

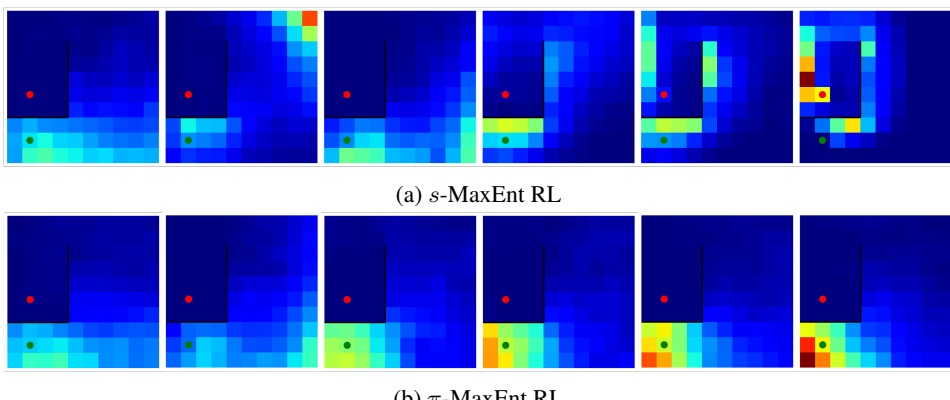

(a) $s$-MaxEnt RL

(b) $\pi$-MaxEnt RL

Figure 1: Comparison of the evolution state-visitation distributions with training for $\pi$-MaxEnt RL and $s$-MaxEnt RL. The darker regions imply lower visitation while the bright regions imply higher visitations.

Since the agent's state visitation distribution depends on both the policy and the dynamics, simply increasing the entropy of the policy (without considering the dynamics) will not ensure that the agent will visit most of the states or will have a state-visitation distribution with high entropy. In Figure 1, we compare the efficiencies of $\pi$-MaxEnt RL and $s$-MaxEnt RL to explore around a wall in a discrete gridworld. The initial and the goal distributions ( highlighted in green and red respectively)

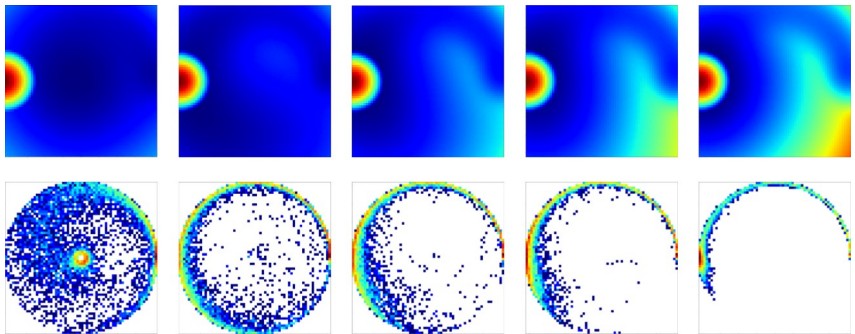

Figure 2: Evolution of $-f'(\frac{p_\theta(s)}{p_g(s)})$ for $f = u \log u$ through policy learning. Top: $f'(\frac{p_\theta(s)}{p_g(s)})$, darker blue are relatively lower values (higher values for the learning signal) while red corresponds to high values (lower values for the learning signal). Bottom: Corresponding state-visitation of the policy.

are separated by a wall. This environment is further discussed in Section 6.1 and Appendix C. Figure 1 shows the evolution of the agent's state-visitation distribution with training for $s$-MaxEnt RL ($fkl$-PG) and $\pi$-MaxEnt RL (Soft Q Learning (Haarnoja et al., 2017))

**Metric-based Shaping Reward:** A deeper look into Lemma 5.1 shows that an appropriate choice of $p_g(s)$ can lead to entropy maximizing policy optimization with metric-based shaping rewards. Define the goal distribution as $p_g(s) = e^{f(s;g)}$ where $f(s;g)$ captures the metric of the underlying space. Then the $fkl$-PG objective becomes,

$$\min D_{FKL}(p_\theta, p_g) = \max \mathbb{E}_{p_\theta}[f(s;g)] - \mathbb{E}_{p_\theta}[\log p_\theta]. \tag{7}$$

The above objective maximizes the reward $f(s;g)$ along with the entropy of the agent's state visitation distribution. For an L2 Euclidean metric, $f(s;g)$ will be $-||s - g||_2^2$ which is the L2 shaping reward, and the goal distribution will be Gaussian. If the goal distribution is Laplacian, the corresponding shaping reward will be the L1 norm.

AIM (Durugkar et al., 2021) used a potential-based shaping reward based on a time step quasimetric. If we define $f(s;g)$ as a Lipschitz function for the time step metric maximizing at $s = g$, we end up optimizing for the AIM reward along with maximizing the entropy of the state-visitation distribution.

### 5.2 Analysis of the learning signals

$f$-PG involves a learning signal $f'(\frac{p_\theta(s)}{p_g(s)})$ to weigh the gradients of log probabilities of the policy. Since we are minimizing the objective (in contrast to policy gradients) the visitation will be pushed towards states with lower values of the learning signal. It is thus important to understand how $f'(\frac{p_\theta(s)}{p_g(s)})$ behaves for goal-conditioned RL settings. During the initial stages of training, the agent visits regions with very low $p_g$. For such states, the signal has a higher value compared to the states that have lower $p_\theta$, i.e., the unexplored states. This is because for any convex function $f$, $f'(x)$ is an increasing function, so minimizing $f'(\frac{p_\theta(s)}{p_g(s)})$ will imply minimizing $p_\theta(s)$ for the states with low $p_g(s)$. The only way to do this is to increase the entropy of the state-visitation distribution, directly making the agent explore new states. As long as there is no significant overlap between the two distributions, it will push $p_\theta$ down to a flatter distribution until there is enough overlap with the goal distribution when it will pull back the agent's visitation again to be closer to the goal distribution.

This learning signal should not be confused with reward in reinforcement learning. It is non-stationary and non-Markovian as it depends on the policy. More importantly, we are not maximizing this signal, just using it to weigh the gradients of the policy.

In the following example, we shall use the Reacher environment (Todorov et al., 2012) to illustrate how our learning signal ($f'(\frac{p_\theta(s)}{p_g(s)})$) varies as the agent learns. We will also show how this signal can push for exploration when the agent has not seen the goal yet. Consider the We fix the goal at (-0.21, 0) and show how the learning signal evolves with the policy. While Figure 2 shows the evolution of $-f'(\frac{p_\theta(s)}{p_g(s)})$ (note the negation) for $fkl$-PG, the rest can be found in Appendix D.

The value of $f'(\frac{p_\theta(s)}{p_g(s)})$ is highest where the agent's visitation is high and lower where the agent is not visiting. $f'(\frac{p_\theta(s)}{p_g(s)})$ has the lowest value at the goal. As the policy converges to the optimal policy, the

regions where the state-visitation distribution is considerably high (towards the bottom-right in the figure), the value for $f'(\frac{p_\theta(s)}{p_g(s)})$ decreases for those states (to still push for exploration) but its value at the goal is low enough for the policy to converge.

# 6 Experiments

Our experiments evaluate our new framework ($f$-PG) as an alternative to conventional reward maximization for goal-conditional RL. We pose the following questions:

1. Does $f$-PG provide sufficient signals to explore in otherwise challenging sparse reward settings?
2. How well does our framework perform compared to discriminator-based approaches?
3. Can our framework scale to larger domains with continuous state spaces and randomly generated goals?
4. How do different $f$-divergences affect learning?

The first two questions are answered using a toy gridworld environment. The gridworld has a goal contained in a room which poses a significant exploration challenge. We also show how the dense signal to the gradients of the policy evolves during training on a continuous domain like Reacher. To answer the third question, our framework is compared with several baselines on a 2D Maze solving task (Point Maze). Additionally, we scale to more complex tasks such as FetchReach Plappert et al. (2018) and an exploration-heavy PointMaze.

## 6.1 Gridworld

We use a gridworld environment to compare and visualize the effects of using different shaping rewards for exploration. We discussed this environment briefly in Section 5.1. The task is for the agent to reach the goal contained in a room. The only way to reach the goal is to go around the wall. The task reward is 1 when the agent reaches the room otherwise it is 0. The state space is simply the $(x, y)$ coordinates of the grid and the goal is fixed. A detailed description of the task is provided in Appendix C. Although the environment seems simple, exploration here is very difficult as there is no incentive for the agent to go around the wall.

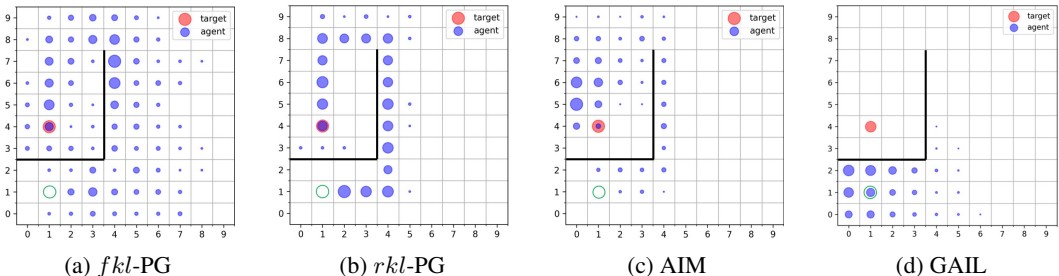

(a) $fkl$-PG      (b) $rkl$-PG      (c) AIM      (d) GAIL

Figure 3: Gridworld: The agent needs to move from the green circle to the red circle. The state visitations of the policies (after 500 policy updates) are shown when using our framework for training (*fkl*, *rkl*) compared with AIM and GAIL trained on top of soft Q learning.

Our framework is compared against AIM (Durugkar et al., 2021), which initially introduced this environment and uses a shaping reward obtained from state-matching to solve it, and GAIL (Ho & Ermon, 2016), which uses a discriminator to learn the probability of a state being the goal state. We provide a comparison to other recent methods in Appendix C. All the baselines are implemented on top of Soft Q Learning (Haarnoja et al., 2017) which along with maximizing the augmented rewards, also maximizes the entropy of the policy while $f$-PG is implemented as an on-policy algorithm without any extrinsic entropy maximization objective. It can be seen from Figure 3 that, $f$-PG can explore enough to find the way around the room which is difficult for methods like GAIL even after the entropy boost. AIM learns a potential function and can also find its way across the wall. As expected, $fkl$-PG converges to the policy maximizing the entropy of the state visitation while $rkl$-PG produces the optimal state visitation as expected from Theorem 5.1. This simple experiment clearly illustrates two things: (1) $f$-PG can generate dense signals to explore the state space and search for the goal and (2) although discriminator-based methods like GAIL try to perform state-matching, they fail to explore the space well.

## 6.2 Point Maze

While the gridworld poses an exploration challenge, the environment is simple and has only one goal. This experiment shows that $f$-PG scales to larger domains with continuous state space and a large set of goals. We use the Point Maze environments (Fu et al., 2020) which are a set of offline RL environments, and modify it to support our online algorithms. The state space is continuous and consists of the position and velocity of the agent and the goal. The action is the force applied in each direction. There are three variations of the environment namely *PointMazeU*, *PointMazeMedium*, *PointMazeLarge*. For the details of the three environments, please refer to Appendix E.

We compare $f$-PG with several goal-based shaping reward, (used alongside the task reward as described in Ng et al. (1999)) to optimize a PPO policy[1]. The rewards tried (along with their abbreviations in the plots) are AIM (Durugkar et al., 2021)(*aim*), GAIL (Ho & Ermon, 2016)(*gail*), AIRL (Fu et al., 2017)(*airl*) and F-AIRL (Ghasemipour et al., 2019)(*fairl*). All these methods employ a state-matching objective. AIM uses Wasserstein's distance while the rest use some form of $f$-divergence. But, all of them rely on discriminators. Along with these baselines, we experiment using our learning signal as a shaping reward (*fkl-rew*). Additionally, we also compare with PPO being optimized by only the task reward (*none*). For our method, we have only shown results for $fkl$-PG. For the rest of the possible $f$-divergences, refer to Section 6.4.

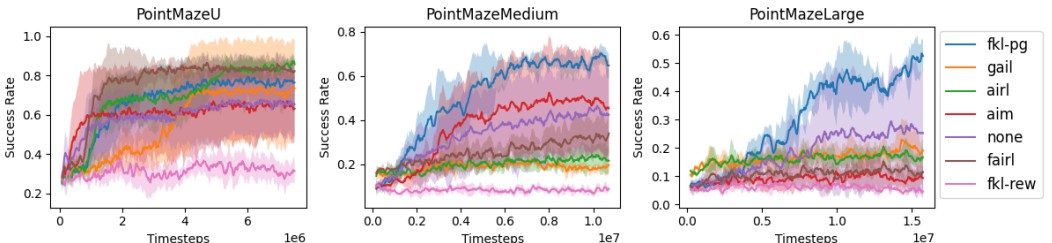

Figure 4: Success rates (averaged over 100 episodes and 3 seeds) of $fkl$-PG and all the baselines. $fkl$-PG performs well in all three environments and better than the baseline shaping rewards in the two tougher environments.

Figure 4 (plotting mean and std-dev for 3 seeds) clearly illustrates that $fkl$-PG is able to perform well in all three environments. In fact, it performs better than the baselines in the more difficult environments. It can also be seen that shaping rewards can often lead to suboptimal performance as *none* is higher than a few of the shaping rewards. As expected, the curve *fkl-new* performs poorly. In the simpler PointMazeU environment, the performance for most of the shaping rewards are similar (along with *none*) but in more complex PointMazeMedium and PointMazeLarge, a lot of these shaping rewards fail.

## 6.3 Scaling to Complex Tasks

We scale our method to more complex tasks such as FetchReach (Plappert et al., 2018) and a difficult version of PointMaze. In the PointMaze environments used in the previous section, distributions from which the initial state and the goal are sampled, have a significant overlap easing the exploration. We modify these environments to ensure a significant distance between the sampled goal distributions and the agent's state-visitation distribution as shown in Figure 5 (top), making exploration highly challenging. Figure 5 (bottom) shows the comparison of $fkl$-PG with GAIL (Ho & Ermon, 2016) and AIM (Durugkar et al., 2021).

The following can be concluded from these experiments: (1) The discriminative-based methods heavily depend on coverage assumptions and fail in situations where there is no significant overlap between the goal distribution and the agent's state visitation distribution. $fkl$-PG does not depend on any such assumptions. (2) $f$-PG is considerably more stable than these baselines (as indicated by the variance of these methods).

## 6.4 Comparing different $f$-divergences

We perform an ablation to compare different $f$-divergences on their performances on the three Point Maze environments. Figure 6 (plotting mean and std-dev for 3 seeds) show that, empirically, $fkl$-PG performs the best followed by $\chi^2$-PG. Interestingly, both of these do not guarantee optimal policies

---

[1]Using spinning up implementation:
https://spinningup.openai.com/en/latest/_modules/spinup/algos/pytorch/ppo/ppo.html

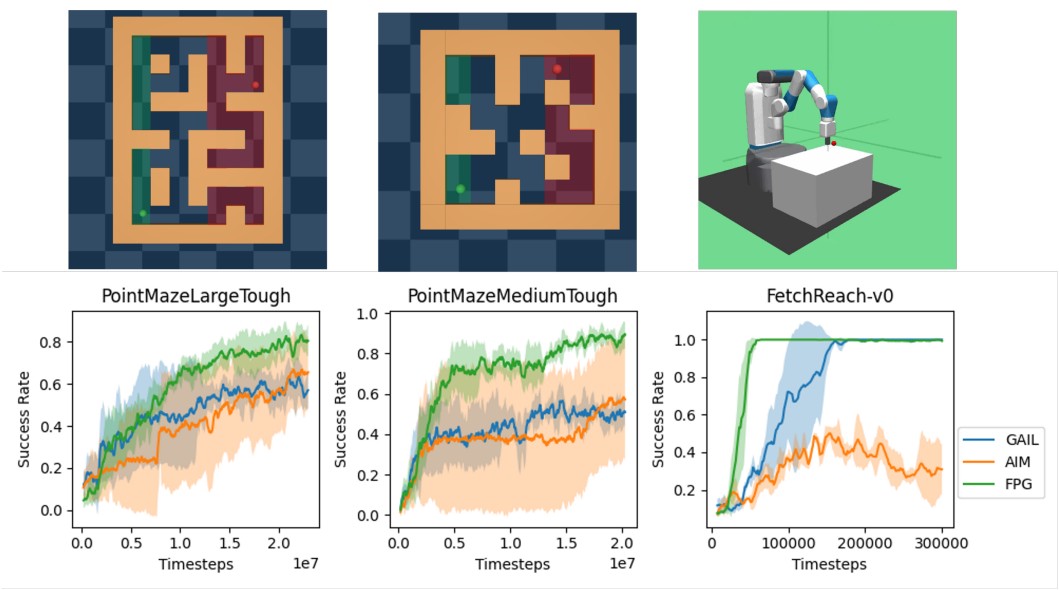

Figure 5: (top): Description of the environments. In the PointMaze environments, the green and red shades represent the distributions from which the initial state and goal states are sampled. (bottom): Success rates (averaged over 100 episodes and 3 seeds) of $fkl$-PG, GAIL and AIM. $fkl$-PG outperforms these baselines with considerably lower variance.

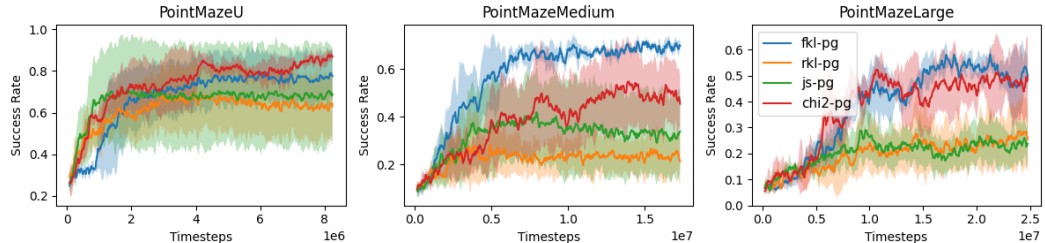

Figure 6: Success rates (averaged over 100 episodes and 3 seeds) of $f$-PG for different $f$. $fkl$-PG performs the best followed by $\chi^2$-PG.

but it can be shown from Lemma 5.1 that $fkl$-PG converges to the policy that along with maximizing for a "reward", maximizes the entropy of the state-visitation. A similar result can be shown for $\chi^2$ as well (proof in the Appendix A). This result can be explained by the need for exploration in the larger mazes, hence learning policies to keep the entropy of the state visitation high.

## 7 Discussion

This paper derives a novel framework for goal-conditioned RL in the form of an on-policy algorithm $f$-policy gradients which minimizes the $f$-divergence between the agent's state visitation and the goal distribution. It proves that for certain $f$-divergences, we can recover the optimal policy while for some, we obtain a policy maximizing the entropy of the state-visitation. Entropy-regularized policy optimization ($s$-MaxEnt RL) for metric-based shaping rewards can be shown as a special case of $f$-PG where $f$ is $fkl$. $f$-PG can provide an exploration bonus when the agent has yet not seen the goal. We demonstrated that $f$-PG can scale up to complex domains.

Through this work, we introduce a new perspective for goal-conditioned RL. By circumventing rewards, $f$-PG can avoid issues that arise with reward misspecification (Knox et al., 2021). There are several avenues to focus on for future work. First, the current framework is on-policy and poses an exploration challenge. An avenue for future work could be to develop an off-policy way to solve the objective. Second, this paper does not tackle goal distributions with several modes. Such a target distribution would be interesting to tackle in future work.

# 8 Acknowledgements

This work was in part supported by Cisco Research. Any opinions, findings and conclusions, or recommendations expressed in this material are those of the authors and do not necessarily reflect the views of Cisco Research.

This work has partially taken place in the Learning Agents Research Group (LARG) at UT Austin. LARG research is supported in part by NSF (FAIN-2019844, NRT-2125858), ONR (N00014-18-2243), ARO (E2061621), Bosch, Lockheed Martin, and UT Austin's Good Systems grand challenge. Peter Stone serves as the Executive Director of Sony AI America and receives financial compensation for this work. The terms of this arrangement have been reviewed and approved by the University of Texas at Austin in accordance with its policy on objectivity in research.

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

# Appendix

# A  Analysis of $J(\theta)$

In this section, we will present the proofs for all the Lemmas and Theorems stated in Section 5.1.

## A.1  Proof for Theorem 5.1

To prove Theorem 5.1, we need the following Lemmas. Lemma A.1 states that among all policies, the optimal policy has the highest state visitation at the goal.

**Lemma A.1.** *Let $\mathcal{D}$ be the set of all possible state visitations for the agent following some policy $\pi \in \Pi$. Let $\pi^*$ be the optimal goal-conditioned policy. This optimal policy's state-visitation distribution will have the most measure at the goal for all $p_\pi \in \mathcal{D}$ i.e., $\pi^* \implies p_{\pi^*}(g) \geq p_\pi(g), \forall p_\pi \in \mathcal{D}$.*

*Proof.* Let $\pi^*$ be the optimal policy and $p_{\pi^*}$ be the corresponding state visitation distribution. The reward for the sparse setting is designed as,

$$r(s) = \begin{cases} 1 & s = g, \\ 0 & \text{otherwise.} \end{cases}$$

Hence the expected return for a policy $\pi$ is $R_\pi$ is

$$R_\pi = \mathbb{E}_{p_\pi}[r(s)]$$
$$= p_\pi(g).$$

The return for the optimal policy is maximum among all policies so $R_{\pi^*} \geq R_\pi, \forall \pi \in \Pi$. This implies $p_{\pi^*}(g) \geq p_\pi(g), \forall p_\pi \in \mathcal{D}$. $\qquad\square$

Lemma A.2 states that the f-divergence between $p_\pi(s)$ and $p_g(s)$ is a decreasing function with respect to $p_\pi(s)$. This means that as the objective $J(\theta)$ obtains its minimum value when $p_\pi(g)$ is highest.

**Lemma A.2.** *$D_f(p_\pi(\cdot)||p_g(\cdot))$ is a decreasing function with respect $p_\pi(g) \forall f$ if $f'(\infty)$ is defined.*

*Proof.* The goal distribution is assumed to be a Dirac distribution i.e., $p_g(s) = 1$ if $s = g$ and 0 everywhere else. The $f$-divergence between the agent state-visitation distribution, $p_\pi$ and the goal distribution, $p_g$ can be defined as,

$$D_f(p_\pi||p_g) = \sum_{p_g > 0} \left[ p_g(s) f\left(\frac{p_\pi(s)}{p_g(s)}\right) \right] + f'(\infty) p_\pi[p_g = 0]$$

$$= f(p_\pi(g)) + f'(\infty)(1 - p_\pi(g)).$$

Let $\mathcal{F} = D_f(p_\pi||p_g)$. Differentiating $\mathcal{F}$ w.r.t. $p_\pi(g)$, we get $\mathcal{F}' = f'(p_\pi(g)) - f'(\infty)$. Since $f$ is a convex function (by the definition of $f$-divergence), $f'(x) \leq f'(y), \forall x \leq y$.

Hence, if $f'(\infty)$ is defined, $\mathcal{F}' \leq 0$. Hence $\mathcal{F} = D_f(p_\pi||p_g)$ is a decreasing function with respect $p_\pi(g)$. $\qquad\square$

Additionally, we need Lemma A.3 and Corollary 1 to complete the proof of Theorem 5.1.

**Lemma A.3.** *If any two policies $\pi_1$ and $\pi_2$ have the same state visitation at a given goal, they have the same returns for that goal.*

*Proof.* Follows directly from the definition of returns. $R_\pi = \mathbb{E}_{p_\pi}[r(s)] = p_\pi(g)$. Hence two policies $\pi_1$ and $\pi_2$ with the same state visitation at the goal will have the same returns. $\qquad\square$

**Corollary 1.** *Any policy that can lead to the state-visitation distribution of the optimal policy $p_{\pi^*}$ is optimal.*

*Proof.* Directly follows from Lemma A.3. $\qquad\square$

**Theorem 5.1.** *The policy that minimizes $D_f(p_\pi||p_g)$ for a convex function $f$ with $f(1) = 0$ and $f'(\infty)$ being defined, is the optimal policy.*

*Proof.* Lemma A.1 proves that the optimal policy has the maximum state-visitation probability. Lemma A.2 proves that the $f$-divergence objective decreases with increasing the state-visitation probability at the goal. In other words, to minimize the $f$-divergence, we need to maximize the state visitation at goal. Corollary 1 further indicates that any policy that can lead to the state-visitation distribution of the optimal policy i.e., any policy that maximizes the state-visitation distribution at the goal state is an optimal policy. $\square$

## A.2 Proof for Lemma 5.1

**Lemma 5.1.** *$fkl$-PG produces a policy that maximizes the reward $\log p_g(s)$ along with the entropy of the state-visitation distribution.*

*Proof.* For $fkl$-PG, $f = u \log u$. Hence, $J(\theta) = D_f(p_\pi||p_g)$ can be written as,

$$
\begin{aligned}
D_f(p_\pi||p_g) &= \mathbb{E}_{p_\pi}\left[\log \frac{p_\pi}{p_g}\right] \\
&= -\left[\mathbb{E}_{p_\pi}[\log p_g] - \mathbb{E}_{p_\pi}[\log p_\pi]\right] \\
&= -\left[\mathbb{E}_{p_\pi}[\log p_g] + \mathcal{H}(p_\pi)\right]
\end{aligned}
$$

where $\mathcal{H}(p_\pi)$ is the entropy of the agent's state visitation distribution. Minimizing $D_f(p_\pi||p_g)$ will correspond to maximizing the reward $r(s) = \log p_g(s)$ and the entropy of $p_\pi$. $\square$

A similar result could be proved for $\chi^2$ divergence:

**Lemma A.4.** *If $f(u) = (u-1)^2$ ($\chi^2$ divergence), $D_f(p_\pi||p_g)$ is the upper bound of $D_{FKL}(p_\pi||p_g) - 1$. Hence minimizing $D_{\chi^2}$ will also minimize $D_{FKL}$ recovering the entropy regularized policy.*

*Proof.* With $f = (u-1)^2$, $D_f(p_\pi||p_g)$ can be written as,

$$
\begin{aligned}
D_f(p_\pi||p_g) &= \int p_g(s)\left(\frac{p_\pi(s)}{p_g(s)} - 1\right)^2 ds \\
&= \int p_g(s)\left(\left(\frac{p_\pi(s)}{p_g(s)}\right)^2 - 2\frac{p_\pi(s)}{p_g(s)} + 1\right)ds \\
&= \int p_\pi(s)\frac{p_\pi(s)}{p_g(s)} - 2p_\pi(s) + p_g(s)ds \\
&= \int p_\pi(s)\frac{p_\pi(s)}{p_g(s)}ds - 1 \\
&= \mathbb{E}_{p_\pi(s)}\left[\frac{p_\pi(s)}{p_g(s)}\right] - 1
\end{aligned}
$$

Since, $x > \log x$,

$$
\begin{aligned}
\implies & \mathbb{E}_{p_\pi(s)}[x] > \mathbb{E}_{p_\pi(s)}[\log x] \\
\implies & \mathbb{E}_{p_\pi(s)}\left[\frac{p_\pi(s)}{p_g(s)}\right] > \mathbb{E}_{p_\pi(s)}\left[\log \frac{p_\pi(s)}{p_g(s)}\right] \\
\implies & \mathbb{E}_{p_\pi(s)}\left[\frac{p_\pi(s)}{p_g(s)}\right] - 1 > \mathbb{E}_{p_\pi(s)}\left[\log \frac{p_\pi(s)}{p_g(s)}\right] - 1
\end{aligned}
$$

Minimizing LHS will also minimize RHS. RHS is essentially $D_{KL}(p_\pi||p_g) - 1$. The $-1$ will not have any effect on the minimization of $D_{KL}(p_\pi||p_g)$. $\square$

# B  Gradient based optimization

## B.1  Derivation of gradients

**Theorem 4.1.** *The gradient of $J(\theta)$ as defined in Equation 4 is given by,*

$$\nabla_\theta J(\theta) = \frac{1}{T} \mathbb{E}_{\tau \sim p_\theta(\tau)} \left[ \left[ \sum_{t=1}^T \nabla_\theta \log \pi_\theta(a_t|s_t) \right] \left[ \sum_{t=1}^T f'\left(\frac{p_\theta(s_t)}{p_g(s_t)}\right) \right] \right]. \tag{8}$$

*Proof.* We follow the proof from Ni et al. (2020). Lets start with the state-visitation distribution. In Section 2, it was shown that the state-visitation distribution can be written as,

$$p_\theta(s) \propto \int p(\tau) \Pi_{t=1}^T \pi_\theta(s_t) \eta_\tau(s) d\tau$$

$$\implies p_\theta(s) \propto \int p(\tau) e^{\sum_{t=1}^T \log \pi_\theta(s_t)} \eta_\tau(s) d\tau$$

$$\implies p_\theta(s) = \frac{\int p(\tau) e^{\sum_{t=1}^T \log \pi_\theta(s_t)} \eta_\tau(s) d\tau}{\int \int p(\tau) e^{\sum_{t=1}^T \log \pi_\theta(s_t)} \eta_\tau(s) d\tau ds}$$

$$\implies p_\theta(s) = \frac{\int p(\tau) e^{\sum_{t=1}^T \log \pi_\theta(s_t)} \eta_\tau(s) d\tau}{\int p(\tau) e^{\sum_{t=1}^T \log \pi_\theta(s_t)} \int \eta_\tau(s) ds d\tau}$$

$$\implies p_\theta(s) = \frac{\int p(\tau) e^{\sum_{t=1}^T \log \pi_\theta(s_t)} \eta_\tau(s) d\tau}{T \int p(\tau) e^{\sum_{t=1}^T \log \pi_\theta(s_t)} d\tau}$$

$$\implies p_\theta(s) = \frac{f(s)}{Z}$$

where $f(s) = \int p(\tau) e^{\sum_{t=1}^T \log \pi_\theta(s_t)} \eta_\tau(s) d\tau$ and $Z = T \int p(\tau) e^{\sum_{t=1}^T \log \pi_\theta(s_t)} d\tau$.

Differentiating w.r.t. $\pi_\theta(s^*)$,

$$\frac{df(s)}{d\pi_\theta(s^*)} = \frac{\int p(\tau) e^{\sum_{t=1}^T \log \pi_\theta(s_t)} \eta_\tau(s) \eta_\tau(s^*) d\tau}{\pi_\theta(s^*)}$$

and,

$$\frac{dZ}{d\pi_\theta(s^*)} = \frac{T \int p(\tau) e^{\sum_{t=1}^T \log \pi_\theta(s_t)} \eta_\tau(s^*) d\tau}{\pi_\theta(s^*)}$$

$$= \frac{T f(s^*)}{\pi_\theta(s^*)}$$

Computing $\frac{dp_\theta(s)}{\pi_\theta(s^*)}$ using $\frac{df(s)}{d\pi_\theta(s^*)}$ and $\frac{dZ}{d\pi_\theta(s^*)}$,

$$\frac{dp_\theta(s)}{\pi_\theta(s^*)} = \frac{Z \frac{df(s)}{d\pi_\theta(s^*)} - f(s) \frac{dZ}{d\pi_\theta(s^*)}}{Z^2}$$

$$= \frac{\int p(\tau) e^{\sum_{t=1}^T \log \pi_\theta(s_t)} \eta_\tau(s) \eta_\tau(s^*) d\tau}{Z \pi_\theta(s^*)} - \frac{f(s)}{Z} T \frac{f(s^*)}{Z \pi_\theta(s^*)}$$

$$= \frac{\int p(\tau) e^{\sum_{t=1}^T \log \pi_\theta(s_t)} \eta_\tau(s) \eta_\tau(s^*) d\tau}{Z \pi_\theta(s^*)} - \frac{T}{\pi_\theta(s^*)} p_\theta(s) p_\theta(s^*)$$

Now we can compute $\frac{dp_\theta(s)}{d\theta}$,

$$\frac{dp_\theta(s)}{d\theta} = \int \frac{dp_\theta(s)}{\pi_\theta(s^*)} \frac{d\pi_\theta(s^*)}{d\theta} ds^*$$

$$= \int \left( \frac{\int p(\tau) e^{\sum_{t=1}^T \log \pi_\theta(s_t)} \eta_\tau(s) \eta_\tau(s^*) d\tau}{Z \pi_\theta(s^*)} - \frac{T}{\pi_\theta(s^*)} p_\theta(s) p_\theta(s^*) \right) \frac{d\pi_\theta(s^*)}{d\theta} ds^*$$

$$= \frac{1}{Z} \int \int p(\tau) e^{\sum_{t=1}^{T} \log \pi_\theta(s_t)} \eta_\tau(s) \eta_\tau(s^*) \frac{\nabla_\theta \pi_\theta(s^*)}{\pi_\theta(s^*)} ds^* d\tau - T p_\theta(s) \int p_\theta(s^*) \frac{\nabla_\theta \pi_\theta(s^*)}{\pi_\theta(s^*)} ds^*$$

$$= \frac{1}{Z} \int p(\tau) e^{\sum_{t=1}^{T} \log \pi_\theta(s_t)} \eta_\tau(s) \sum_{t=1}^{T} \nabla_\theta \log \pi_\theta(s_t) d\tau - T p_\theta(s) \mathbb{E}_{s \sim p_\theta(s)}[\nabla_\theta \log \pi_\theta(s)]$$

The objective $L(\theta) = D_f(p_\theta(s) || p_g(s)) = \int p_g(s) f\left(\frac{p_\theta(s)}{p_g(s)}\right) ds$

The gradient for $L(\theta)$ will be given by,

$$\nabla_\theta L(\theta) = \int p_g(s) f'\left(\frac{p_\theta(s)}{p_g(s)}\right)\left(\frac{\nabla_\theta p_\theta(s)}{p_g(s)}\right) ds$$

$$= \int \nabla p_\theta(s) f'\left(\frac{p_\theta(s)}{p_g(s)}\right) ds$$

$$= \int f'\left(\frac{p_\theta(s)}{p_g(s)}\right)\left(\frac{1}{Z}\int p(\tau) e^{\sum_{t=1}^{T} \log \pi_\theta(s_t)} \eta_\tau(s) \sum_{t=1}^{T} \nabla_\theta \log \pi_\theta(s_t) d\tau\right.$$
$$\left. - T p_\theta(s) \mathbb{E}_{s \sim p_\theta(s)}[\nabla_\theta \log \pi_\theta(s)]\right) ds$$

$$= \frac{1}{Z} \int p(\tau) e^{\sum_{t=1}^{T} \log \pi_\theta(s_t)} \sum_{t=1}^{T} \nabla_\theta \log \pi_\theta(s_t) \int \eta_\tau(s) f'\left(\frac{p_\theta(s)}{p_g(s)}\right) ds d\tau$$
$$- \int T p_\theta(s) f'\left(\frac{p_\theta(s)}{p_g(s)}\right) \mathbb{E}_{s \sim p_\theta(s)}[\nabla_\theta \log \pi_\theta(s)] ds$$

$$= \frac{1}{Z} \int p(\tau) e^{\sum_{t=1}^{T} \log \pi_\theta(s_t)} \sum_{t=1}^{T} \nabla_\theta \log \pi_\theta(s_t) \sum_{t=1}^{T} f'\left(\frac{p_\theta(s_t)}{p_g(s_t)}\right) d\tau$$
$$- T \int p_\theta(s) f'\left(\frac{p_\theta(s)}{p_g(s)}\right) \mathbb{E}_{s \sim p_\theta(s)}[\nabla_\theta \log \pi_\theta(s)] ds$$

$$= \frac{1}{T} \int p_\theta(\tau) \sum_{t=1}^{T} \nabla_\theta \log \pi_\theta(s_t) \sum_{t=1}^{T} f'\left(\frac{p_\theta(s_t)}{p_g(s_t)}\right) d\tau$$
$$- T \mathbb{E}_{s \sim p_\theta(s)}\left[f'\left(\frac{p_\theta(s)}{p_g(s)}\right)\right] \mathbb{E}_{s \sim p_\theta(s)}[\nabla_\theta \log \pi_\theta(s)]$$

$$= \frac{1}{T} \mathbb{E}_{\tau \sim p_\theta(\tau)}\left[\sum_{t=1}^{T} \nabla_\theta \log \pi_\theta(s_t) \sum_{t=1}^{T} f'\left(\frac{p_\theta(s_t)}{p_g(s_t)}\right)\right]$$
$$- T \mathbb{E}_{s \sim p_\theta(s)}\left[f'\left(\frac{p_\theta(s)}{p_g(s)}\right)\right] \mathbb{E}_{s \sim p_\theta(s)}[\nabla_\theta \log \pi_\theta(s)]$$

$$= \frac{1}{T}\left[\mathbb{E}_{\tau \sim p_\theta(\tau)}\left[\sum_{t=1}^{T} \nabla_\theta \log \pi_\theta(s_t) \sum_{t=1}^{T} f'\left(\frac{p_\theta(s_t)}{p_g(s_t)}\right)\right]\right.$$
$$\left. - \mathbb{E}_{s \sim p_\theta(s)}\left[\sum_{t=1}^{T} f'\left(\frac{p_\theta(s_t)}{p_g(s_t)}\right)\right] \mathbb{E}_{\tau \sim p_\theta(\tau)}[\sum_{t=1}^{T} \nabla_\theta \log \pi_\theta(s_t)]\right]$$

$$= \frac{1}{T} \mathbb{E}_{\tau \sim p_\theta(\tau)}\left[\sum_{t=1}^{T} \nabla_\theta \log \pi_\theta(s_t) \sum_{t=1}^{T} f'\left(\frac{p_\theta(s_t)}{p_g(s_t)}\right)\right]$$

$\square$

**Theorem 4.2.** *Updating the policy using the gradient maximizes* $\mathbb{E}_{p_\theta}[\eta_\tau(g)]$.

*Proof.* In goal-based setting, $p^g$ is sparse, so we need to use the full definition of $f$-divergence, $D_f(p_\theta||p_g) = \sum_{p_g>0}\left[p_g(s)f(\frac{p_\theta(s)}{p_g(s)})\right] + f'(\infty)p_\theta[p_g = 0] = \sum_{p_g>0}\left[p_g(s)f(\frac{p_\theta(s)}{p_g(s)})\right] + f'(\infty)(1 - p_\theta(g))$. Differentiating with respect to $\theta$ gives,

$$\nabla_\theta L(\theta) = \left(f'(p_\theta(g)) - f'(\infty)\right)\nabla_\theta p_\theta(s)$$

$$= \left(f'(p_\theta(g)) - f'(\infty)\right)\left(\frac{1}{T}\int p_\theta(\tau)\eta_\tau(g)\sum_{t=1}^T \nabla_\theta \log \pi_\theta(s_t)d\tau - Tp_\theta(s)\mathbb{E}_{s\sim p_\theta(s)}[\nabla_\theta log\pi_\theta(s)]\right)$$

$$= \left(f'(p_\theta(g)) - f'(\infty)\right)\left(\frac{1}{T}\mathbb{E}_{\tau\sim p_\theta(\tau)}\left[\eta_\tau(g)\sum_{t=1}^T \nabla_\theta \log \pi_\theta(s_t)\right]\right.$$

$$\left. - p_\theta(g)\mathbb{E}_{\tau\sim p_\theta(\tau)}\left[\sum_{t=1}^T \nabla_\theta \log \pi_\theta(s)\right]\right)$$

$$= \frac{1}{T}\left(f'(p_\theta(g)) - f'(\infty)\right)\mathbb{E}_{\tau\sim p_\theta(\tau)}\left[\sum_{t=1}^T \nabla_\theta \log \pi_\theta(s_t)\eta_\tau(g)\right]$$

The gradient has two terms, the first term $\left(f'(p_\theta(g)) - f'(\infty)\right)$ weighs the gradient based on the value of $p_\theta(g)$ and is always negative. It acts as an adaptive learning schedule, reducing its magnitude as the $p_\theta(g)$ increases. The second term is the gradient of $\mathbb{E}_{p_\theta}[\eta_\tau(g)]$. Hence using $\nabla_\theta L(\theta)$, we minimize $L(\theta)$ which would imply maximizing $\mathbb{E}_{p_\theta}[\eta_\tau(g)]$. $\qquad\square$

## B.2 Practical Algorithm

As mentioned in Section 4, the derived gradient is highly sample inefficient. We employ established methods to improve the performance of policy gradients like importance sampling.

The first modification is to use importance sampling weights to allow sampling from previous policy $\theta'$. The gradient now looks like,

$$\nabla_\theta J(\theta) = \mathbb{E}_{\tau\sim p_{\theta'}(\tau)}\left[\frac{\pi_\theta(\tau)}{\pi_{\theta'}(\tau)}\left[\sum_{t=1}^T \nabla_\theta \log \pi_\theta(a_t|s_t)\right]\left[\sum_{t=1}^T f'\left(\frac{p_\theta(s_t)}{p_g(s_t)}\right)\right]\right]. \tag{9}$$

To reduce the variance in the gradients, the objective can be modified to use the causal connections in the MDP and ensure that the action taken at step $t$ only affects rewards at times $t' \to [t, T]$. Moreover, a discount factor $\gamma$ is used to prevent the sum $\sum_{t'=t}^T f'\left(\frac{p_\theta(s_t)}{p_g(s_t)}\right)$ from exploding.

Additionally, the expectation is modified to be over states rather than trajectories,

$$\nabla_\theta J(\theta) = \mathbb{E}_{s_t,a_t\sim p_{\theta'}(s_t,a_t)}\left[\frac{\pi_\theta(a_t|s_t)}{\pi_{\theta'}(a_t|s_t)}\nabla_\theta \log \pi_\theta(a_t|s_t)\sum_{t'=t}^T \gamma^{t'} f'\left(\frac{p_\theta(s_t)}{p_g(s_t)}\right)\right]. \tag{10}$$

This gradient computation is still inefficient, because even though the samples are from a previous policy $\pi_{\theta'}$, it still needs to compute $\sum_{t'=t}^T \gamma^{t'} f'\left(\frac{p_\theta(s_t)}{p_g(s_t)}\right)$, requiring iteration through full trajectories.

We can add a bias to the gradient by modifying $f'\left(\frac{p_\theta(s_t)}{p_g(s_t)}\right)$ to $f'\left(\frac{p_{\theta'}(s_t)}{p_g(s_t)}\right)$ in the objective. To ensure the bias is small, an additional constraint needs to be added to keep $\theta'$ close to $\theta$. Following the literature from natural gradients, the constraint we add is $D_{KL}(p_{\theta'}||p_\theta)$. Proximal Policy Optimization (Schulman et al., 2017) showed that in practical scenarios, clipped objective can be enough to do away with the KL regularization term. The final objective that we use is,

$$\nabla_\theta J(\theta) = \mathbb{E}_{s_t,a_t\sim p_{\theta'}(s_t,a_t)}\left[\min(r_\theta(s_t)F_{\theta'}(s_t), clip(r_\theta(s_t), 1-\epsilon, 1+\epsilon)F_{\theta'}(s_t))\right], \tag{11}$$

where $r_\theta(s_t) = \frac{\nabla_\theta \pi_\theta(a_t|s_t)}{\pi_{\theta'}(s_at|s_t)}$ and $F_{\theta'}(s_t) = \sum_{t'=t}^T \gamma^{t'} f'\left(\frac{p_{\theta'}(s_t)}{p_g(s_t)}\right)$.

## B.3 Discounted State-Visitations

The state-visitation distribution defined so far has not considered a discount factor. To include discounting, the state-visitation frequency gets modified to $\eta_\tau(s) = \sum_{t=1}^T \gamma^t \mathbb{1}_{s_t=s}$. Throughout the derivation of the gradient, we used $\int \eta_\tau(s)f(s)ds = \sum_{t=1}^T f(s_t)$ but this will be modified to $\int \eta_\tau(s)f(s)ds = \sum_{t=1}^T \gamma^t f(s_t)$. the corresponding gradient will be,

$$\nabla_\theta J(\theta) = \frac{1}{T}\mathbb{E}_{\tau\sim p_\theta(\tau)}\left[\left[\sum_{t=1}^T \gamma^t \nabla_\theta \log \pi_\theta(a_t|s_t)\right]\left[\sum_{t=1}^T \gamma^t f'\left(\frac{p_\theta(s_t)}{p_g(s_t)}\right)\right]\right]. \quad (12)$$

This gradient can be modified as before,

$$\nabla_\theta J(\theta) = \mathbb{E}_{s_t,a_t\sim p_\theta(s_t,a_t)}\left[\gamma^t\nabla_\theta \log \pi_\theta(a_t|s_t)\sum_{t'=t}^T \gamma^{t'} f'\left(\frac{p_\theta(s_t)}{p_g(s_t)}\right)\right]. \quad (13)$$

Adding importance sampling to the gradient in Equation 13 will give a gradient very similar to Equation 10. In fact, Equation 10 is a biased estimate for the gradient of the $f$-divergence between the discounted state-visitation distribution and the goal distribution. We can use either of the two gradients but Equation 10 will be preferred for long horizon tasks.

## C Gridworld Experiments

### C.1 Description of the task

The task involves navigating a gridworld to reach the goal state which is enclosed in a room. The agent can move in any of the four directions and has no idea where the goal is. It needs to explore the gridworld to find the path around the room to reach the goal. The task is further elaborated in Figure 7. The green square represents the agent position while the red square represents the goal.

**State:** The state visible to the policy is simply the normalized $x$ and $y$ coordinates.

**Action:** The action is discrete categorical distribution with four categories one for each - left, top, right and bottom.

**Reward:** The task reward is 1 at the goal and 0 everywhere else. $f$-PG does not require rewards but the baselines use task rewards.

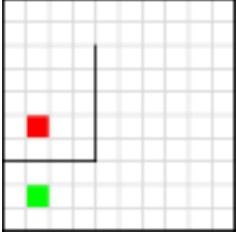

Figure 7: Description of the gridworld: The bold lines show the walls, green square is the start position and the red square is the goal.

### C.2 Performance of $f$-PG

In Section 5.1, we had compared $fkl$-PG and $rkl$-PG with AIM (Durugkar et al., 2021) and GAIL (Ho & Ermon, 2016). In Figure 8 we present additional baselines AIRL (Fu et al., 2017) and f-AIRL (Ghasemipour et al., 2019).

## D Visualizing the learning signals

### D.1 Description of the task

To visualize the learning signals, we use the Reacher environment (Figure 9) (Todorov et al., 2012). The task involves rotating a reacher arm (two joints with one end fixed). The applied actions (torques) would rotate the arm so that the free end reaches the goal. The goal is fixed to be at $(-0.21, 0)$ and the goal distribution is a normal centred at the goal with a standard deviation of 0.02.

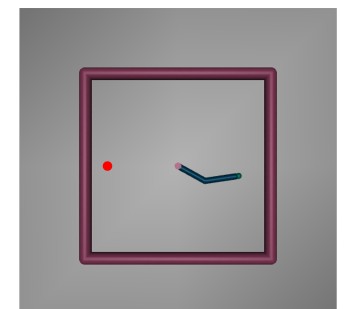

Figure 9: The Reacher environment with fixed goal.

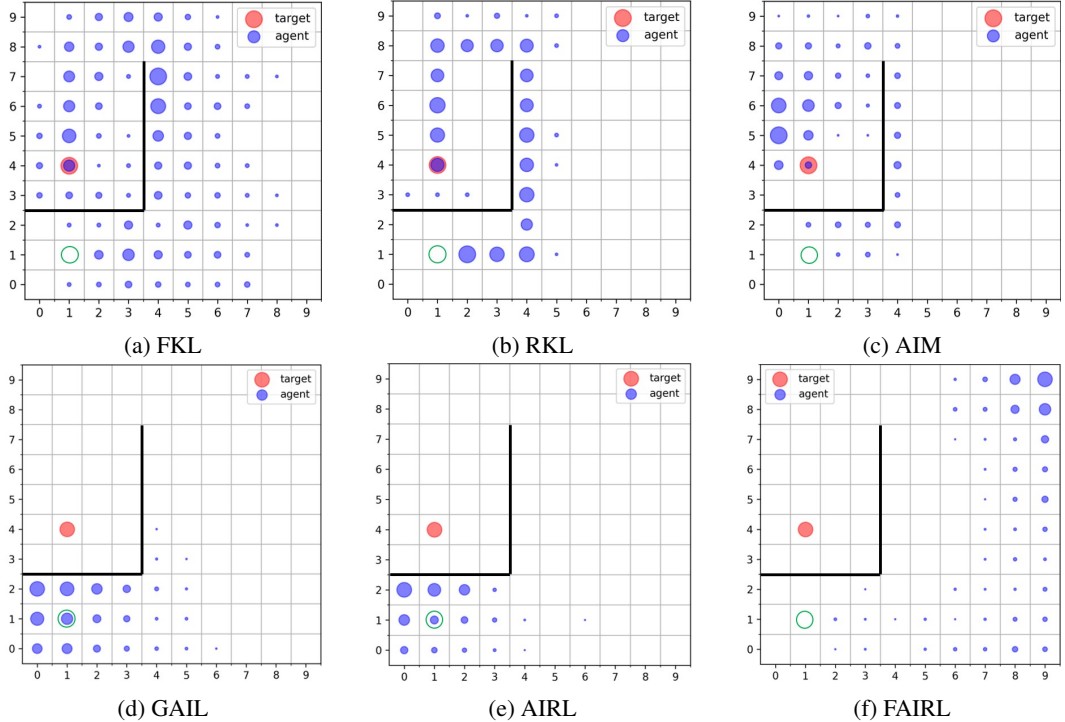

Figure 8: Gridworld: The agent needs to move from the green circle to the red circle. The state visitations of the final policies are shown when using our framework for training (*fkl*, *rkl*) compared with AIM and GAIL trained on top of soft Q learning.

**State:** The state of the original environment contains several things but here we simplify the state space to simply be the position of the free end and the target or the goal position.

**Actions:** The actions are two dimensional real numbers in $[-1, 1]$ which correspond to the torques applied on the two joint respectively.

**Reward:** The reward is sparse i.e., 1 when the goal is reached by the tip of the arm. But $f$-PG does not use rewards for training policies.

## D.2 Comparing different $f$-PG

Figure 10 shows the evolution of $-f'\left(\frac{p_\theta(s)}{p_g(s)}\right)$ for the environment. The *red* regions correspond to signals having a lower value while the darker *blue* regions correspond to signals with high value. For $fkl$, the scale of these signals generally varies from 10 to $-5$ while for $\chi^2$, the scale varies from 600 to 50. Also, for the same objective, as the policy trains, these scales generally get smaller in magnitude.

The following can be observed from these plots:

1. In all the cases, the signals are maximum at the the goal pulling the state-visitations towards the goal.

2. All of these also push for exploration. This is most pronounced in $fkl$ and $\chi^2$. These provide significant push towards the unexplored regions which show their inclination towards entropy-maximization, confirming the theory (Lemma 5.1).

# E  PointMaze experiments

PointMaze (Fu et al., 2020) are continuous state-space domains where the agent needs to navigate to the goal in the 2D maze. The agent and the goal are spawned at a random location in the maze for every episode. There are three levels based on the difficulty of the maze as shown in Figure 11.

**State:** The state consists of the agent's 2D position and the velocity in the maze. The goal position is appended to the state.

**Action:** The actions are 2D real numbers in the range $[-1, 1]$ correspond to the force applied to the agent in each of the two directions.

**Reward:** Although $f$-PG does not use rewards, the baselines use the task reward which is sparse (1 when the goal is reached and 0 everywhere else).

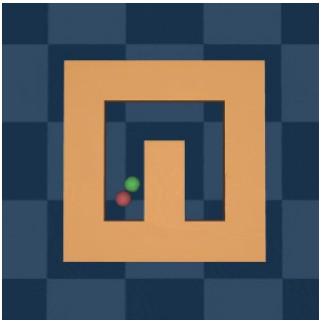 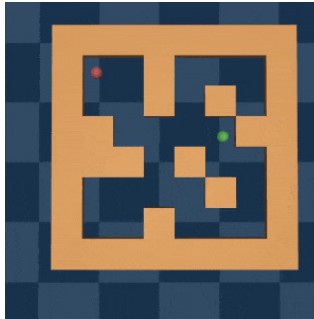 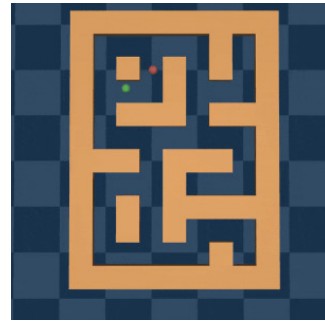

Figure 11: Description of PointMaze environments: PointMaze-U (left), PointMaze-Medium (middle), PointMaze-Large(right).

For the experiments in Section 6.2, the initial and goal states are sampled uniformly over all the "VALID" states i.e., states that can be reached by the agent. Such an initialization allows discriminator-based methods to fulfill their coverage assumptions. In Section 6.3, the initialization procedure is modified so that the initial state and the goal state are considerably far. This is done by restricting the sampling of the initial and goal states from disjoint (considerably far away) distributions as shown in Figure 5.

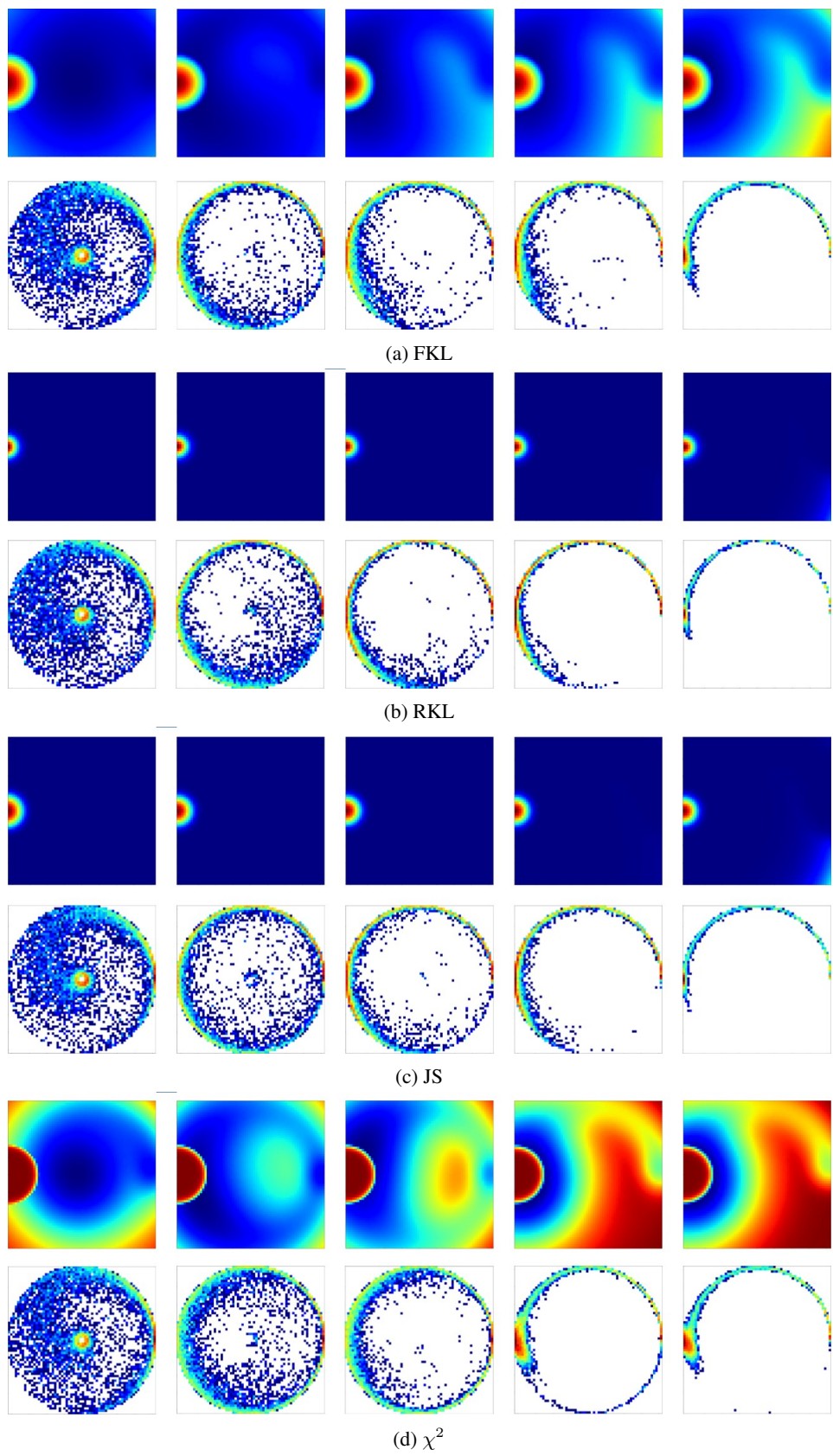

Figure 10: Evolution of $-f'\left(\frac{p_\theta(s)}{p_g(s)}\right)$ along with the corresponding state-visitations for different $f$-divergences - $fkl$, $rkl$, $js$ and $\chi^2$. The scales for the value of these signals are not shown but they vary as the policy converges. $f$-PG provides dense signals for pushing towards exploration.

