# OpenReview forum: "f-Policy Gradients: A General Framework for Goal-Conditioned RL using f-Divergences"
_NeurIPS.cc/2023/Conference — NeurIPS 2023 poster_

### Official Review · Reviewer_trTZ · 2023-06-20

**Soundness:** 3 good
**Presentation:** 4 excellent
**Contribution:** 2 fair
**Rating:** 4
**Confidence:** 4

**Summary:**

This paper studies goal-conditioned reinforcement learning and proposes a framework based on f-divergence. The authors show that minimizing the f-divergence between the agent’s visitation and the proposed goal distribution leads to the optimal policy and discuss the practical implementations, connections with previous GCRL methods.

**Strengths:**

1. The paper is well written and easy to follow.
2. The motivation to study the divergence between the state visitation and goal distribution is straightforward, which is good.
3. The analysis is thorough. The discussion between the proposed framework and previous methods is also interesting.

**Weaknesses:**

1. The authors state that they are focusing on sparse-reward (1 if the agent reaches the goal and 0 otherwise). However, $p_g(s)$ will become a Dirac distribution and the f-policy gradient objective makes no sense in this case.
2. As the authors admit, the algorithm and analysis are meaningless in a full Dirac distribution setting. The authors choose to "add an ϵ probability to all other states". Will this uniform ϵ addition + f-PG performs better than uniform reward shaping + standard PG with GC-reward? Intuitively, such uniform addition will still lead to sparse training signal.
3. It is not very clear about the advantage of the proposed framework. In the strict sparse-reward setting, the framework degrades. Even in dense-reward settings, the analysis only states that optimizing the f-PG objective moves to the optimal policy, which is also true for standard PG methods in GCRL that are based on goal-conditioned (dense) rewards only. Are there any efficiency advantage of f-PG?

**Questions:**

See the weakness section above.

**Limitations:**

The authors only discussed some future avenues of their work. I suggest the authors to discuss more about the benefit over the shaping rewards GCRL algorithms, and discuss the previous theoretical results established for shaping rewards methods.

---

> ### Author Rebuttal · Authors · 2023-08-09
>
> We thank the reviewers for lending their expertise to reviewing our paper, as well as for their constructive critique and suggestions for improvement.
>
> We have provided a detailed analysis of the use of the Dirac distribution in our common rebuttal. We request the reviewer to go refer to it if clarifications are required.
>
> Weaknesses:
>
> (1) The f-divergence objective (Equation 4) works for both situations when the goal distribution is sparse or dense. Section 4.1 provides the analysis assuming that the goal distribution is sparse. The gradient in Theorem 4.2 is not defined when the p_g(s) is sparse but the gradient derived for such cases is provided in Theorem 4.3. Subsequent paragraphs explain how to use the gradient from Theorem 4.2 in sparse reward settings.
>
> (2) In all the experiments, the goal distribution is sparse with an $\epsilon$ approximation. In all cases, it performs better than uniform shaping rewards.
>
> (3) All our experiments are in sparse settings where our method outperforms the standard PG methods. We show that in sparse reward schemes, we are able to provide dense signals for training. These dense signals are different from dense rewards. For each f-divergence, we get a different type of dense learning signal. In the Reacher environment, we show how these learning signals evolve with training in the supplementary section for different f-divergences. While in Section 5.3, we do show these dense signals for one particular f-divergence.
>
> Limitations:
>
> We have discussed connections to metric based shaping rewards in Section 4.4 and how it is better as it encourages exploration.

---

> > ### Comment · Reviewer_trTZ · 2023-08-17
> >
> > Thank the authors for the response. My concern regarding the advantage of f-PG compared to standard GCRL still remains. The authors provide experimental results that compare f-PG and GCRL, but didn't give enough incentive to choose f-PG from a theoretical perspective. Can the authors explain in more detail? \
> > Besides, can the authors also comment on what is the connection and difference between f-PG and GoFAR? I understand that GoFAR works in offline setting, but the formula and the original objective is pretty similar. Is it possible to generalize f-PG to offline settings?

---

> > > ### Author Response · Authors · 2023-08-17
> > > **Clarifications to the theoretical incentive for f-PG and differences with GoFAR**
> > >
> > > Thank you for your response.
> > >
> > > We would like to address both your concerns:
> > >
> > > (1) Regarding the theoretical incentive: Most GCRL algorithms rely on learnt shaping rewards. A majority of these use adversarial training involving a discriminator which is not stable. Mathematically, these rewards are not stationary, and depend on the policy, so while policy optimization, they shouldn't be treated as constant rewards - which these works ignore. Moreover, discriminators assume enough coverage between the agent's state visitation and the goal distribution. f-PG follows a mathematically principled approach for minimizing f-divergence. The learning signals that we obtain are dense, incentivizes exploration and provably produces optimal policies.
> > >
> > > (2) Differences with GoFAR: GoFAR aims to minimize the KL Divergence between the agent's state visitation distribution and the goal distribution but it creates a lower bound where it maximizes a reward of $\log{\frac{p(s;g)}{d^O(s;g)}}$, where $p(s;g)$ is the goal distribution and $d^O(s;g)$ is the offline state visitation distribution. This reward is learnt from a discriminator. The f-divergence in GoFAR is used to regularize the policy to be close to the offline dataset. Our work directly minimizes the f-divergence (not specifically only KL) between the agent's visitation and the goal distribution. The rewards used by GoFAR do not depend on the policy but the learning signals that we use constantly evolve with the policy. Moreover, there is no incentive for exploration in the reward used by GoFAR (the incentive is to increase the state visitation at "expert" states compared to their corresponding visitations in the offline dataset) and during training it assumes $d^O(s;g)$ to be non zero at the goal (coverage assumption). f-PG generates learning signals that incentivizes exploration and we do not assume coverage.
> > >
> > > Our method is a completely on-policy as we have discussed in the limitations. Currently, f-PG cannot be adapted directly into an offline framework. Although we will be exploring ways to do so in future works.

---

> > > > ### Author Response · Authors · 2023-08-20
> > > >
> > > > We appreciate the reviewer trTZ for their devoting time for the review and providing valuable suggestions. We hope that we have addressed all the concerns of the reviewer. We would like to reiterate our clarifications:
> > > >
> > > > The theoretical incentive for using f-PG is:
> > > >
> > > > (1) A mathematically stable objective that does not use an algorithm that assumes stationary reward functions over non-stationary ones. Even works like C-Learning [1] that do not use discriminators make similar assumptions while policy optimization.
> > > >
> > > > (2) Obtain learning signals that incentivise exploration and provably produces optimal policies without any coverage assumptions.
> > > >
> > > > While GoFAR aims at minimizing the KL-divergence between agent’s state visitation distribution and the goal distribution,
> > > >
> > > > (1) We derive our results for a general f-divergence rather than using a variational lower bound that involves training a discriminator.
> > > >
> > > > (2) Our learning signals incentivize exploration and do not make any coverage assumptions while GoFAR does not provide any incentive for exploration. The rewards used by GoFAR only pushes the distribution of the expert states to be higher than that in the offline dataset.
> > > >
> > > > [1] B. Eysenbach, et. al., C-Learning: Learning to Achieve Goals via Recursive Classification, ICLR 2021

---

### Official Review · Reviewer_PJFh · 2023-06-27

**Soundness:** 2 fair
**Presentation:** 3 good
**Contribution:** 3 good
**Rating:** 6
**Confidence:** 3

**Summary:**

The paper focuses on goal-conditioned reinforcement learning (RL), where the goal-conditioned reward function is often sparse. Consequently, standard goal-conditioned RL algorithms suffer from the exploration problem, thus requiring the need to shape rewards. The paper proposes a a new formulation that minimizes the $f$-divergence between the policy's state-visitation distribution and the goal distribution. The paper theoretically shows that minimizing such objective yields the optimal policy and provided empirical evaluations on two maze tasks.

**Strengths:**

- As far as I know, the $f$-divergence between state-visitation distribution and goal distribution is a novel contribution.
- The paper theoretically shows that optimizing their objective yields the optimal policy.
- Good visualizations on demonstrating the policies' behaviours.

**Weaknesses:**

- I think $\gamma$ should not be introduced as it does not really contribute to the idea. Also, since we wish to maximize $\mathbb{E}_\theta\left[\eta\right]$, the aim to achieve the task as quickly as possible is there.
- How is $\int \eta_\tau(s) ds = T$ in theorem 4.3 when we are not assuming fixed horizon $T$---this is based on page 3, on line 131, where the paper assumes that the trajectory ends when the agent reaches the goal or if maximum horizon $T$ is reached.
	- It seems like if the paper is using the fixed-horizon setting, then theorem 4.3 will make sense, in terms of being similar to maximizing the expected return.
	- Furthermore, I believe we can actually relate the policy error even if the trained policy is sub-optimal. It will be nice if there is some results regarding suboptimality.
- On page 5, second last paragraph, why do we not just assign equiprobability to the correct goal states, but $\varepsilon$ to all other states? It seems like we already have such access anyway. Otherwise, is this not a "coverage" assumption?
- If I am not mistaken, under Gaussian policies, $f$-PG with FKL essentially recovers some sort of L2 distance. I think for reaching tasks this is naturally a good and dense reward, which may explain the better performance than other approaches.
	- It will be interesting if the paper provides plot describing the correlation of the "constructed" reward and the "natural" dense reward (i.e. L2 distance between state and goal).
	- It will also be nice to show a task aside from maze and analyze whether FKL still performs better in general.
- The paper should further describe how the state-visitation distribution $p_\pi$ and the goal distribution $p_g$ are trained, maybe in the appendix.

**Questions:**

- Based on Figure 4, the paper suggests that the maximizing entropy of state-visitation matters.
	- When computing the success rate, are we sampling from the policy, or taking the mean of the policy?
	- How would PPO with entropy-regularized objective (or SAC) perform?
- The paper proposes to learn the distributions separately using kernel density estimators (KDE).
	- It seems like this is not really the consequence of the theory, but just a choice, is that right?
	- If it is indeed a choice, I am not completely sure if the instability of the discriminator plays a role.
	- In the related works section, there were few sentences indicating instability due to non-stationary rewards (in particular, under the subsection shaping rewards). Isn't this approach also non-stationary? Shouldn't there be similar instability as other approaches then?

**Possible typos**
- On page 2, line 52: "insufficiency" instead of "insufficient"
- On page 2, line 90: "learning" instead of "Learning"
- On page 3, line 94: "expert's" instead of "expert"
- On page 3, line 102: "learning" instead of "Learning"
- On page 3, line 133: The expectation should also be over the goal $g$
- On page 3, line 137: The distribution should include $\mu_0(s_0)$
- On page 3, equation (1): $p_\theta(\tau ; g)$ instead of $p_\theta(\tau)$
- On page 3, equation (2): $(s) d\tau$ should be $d\tau$ on the numerator
- On page 4, line 160: $\pi$ instead of $p_\pi$.
- Figure 2, legend: $fkl-rew$ should be $fkl-new$
- Appendix, on page 6, line 86: Should be "Theorem 4.3" instead of "Theorem 4.2"

**Limitations:**

- The paper requires an assumption of knowing the goal function. That is, reward $r(s, a, s';g) = \mathbb{I}(s' = g)$. Designing such a goal function can be challenging due to hardware constraints in real-life applications, notably in robotics tasks.

---

> ### Author Rebuttal · Authors · 2023-08-09
>
> We thank the reviewers for lending their expertise to reviewing our paper, as well as for their constructive critique and suggestions for improvement.
> We have addressed several of the concerns in the main rebuttal as well.
>
> Weaknesses:
>
> (1) $\gamma$ is used later in 4.2 when deriving the practical gradient.
>
> (2) Thank you for pointing this out. We consider the fixed horizon setting. Only during evaluations, the episodes end when the goal is reached. During training, we terminate after a fixed episode length. We will correct this in our camera ready version.
>
> (3) We have not looked into policy errors for suboptimal policies in this work. It is an interesting direction for future research, and we thank the reviewer for suggesting it.. In this work, we focus on proving that our algorithm gives us the optimal policy.
>
> (4) We have discussed the approximation in detail in our main rebuttal. We are providing equiprobability to all the goal states and $\epsilon$ to all the other states. This distribution does not require any coverage assumptions on the part of the agent’s visitation.
>
> (5) A Gaussian policy with f-PG does not lead to an L2, rather assuming a gaussian distribution over the goal with FKL can lead to an L2 like shaping reward but with an additional term for maximizing the entropy of the state-visitation distribution. In the gridworld and maze environments, using only L2 performs very poorly as the agent gets stuck by the walls.
>
> (6) We have discussed the use of KDE in the main rebuttal. We shall provide an algorithm box in our camera ready version.
>
> Questions:
>
> (1) While evaluating we always use the mean of the Gaussian distribution generated by the policy.
>
> (2) As explained in our main rebuttal, FKL is different from MaxEnt RL. Still, in Section 5.1, all the baselines used a policy entropy-maximizing objective (based on MaxEnt RL) which failed to outperform f-PG (detailed results in supplementary material (D.2)).
>
> (3) KDE is a choice and independent of the theory.
>
> (4) In previous works like GAIL, AIRL, f-AIRL, the objective itself is defined as a min-max objective so they cannot do away with the discriminator. Since we do not have a min-max objective, we can do away with the discriminator and its instabilities.
>
> (5) The previous methods perceive their non-stationary rewards as stationary and use conventional RL to maximize the reward. They assume reward is independent of the policy throughout their optimization. We do not  make such an assumption. We obtain a gradient weighted by a signal that depends on the policy. Thus the signal we obtain cannot be considered equivalent to the reward used by previous approaches.
>
> Limitation:
>
> In any continuous domain, the goal distribution can be defined using a normal distribution with very small variance. This is a standard assumption[1, 2] in most of the robotic domains as well. In point maze, we have used the same assumption.
>
> [1] I. Durugkar, et. al., Adversarial Intrinsic Motivation for Reinforcement Learning, NeurIPS 2021
>
> [2] B. Eysenbach, et. al., C-Learning: Learning to Achieve Goals via Recursive Classification, ICLR 2021

---

> > ### Comment · Reviewer_PJFh · 2023-08-15
> >
> > Thank you for the detailed response.
> >
> > Regarding the general response to all reviewers, I have a question on (3) and (4), (i.e. figure 2 in the new PDF.)
> > If I understand correctly, the cell values in the figures 2(c) and 2(d) are the reward functions for policy gradient. In both cases would the agent not try to escape the negative-valued states and stay in the non-negative states? It will also depend on the horizon of the trajectories that allow the agent to trade-off staying vs moving towards the "best" cell.
> >
> > I acknowledge and thank the authors for the responses specific to my previous comments, though I have few follow-up questions:
> >
> > Weaknesses:
> > (1) We can still use policy gradient without discount factor when we are in the fixed-horizon setting, is that not correct? One main point of $\gamma$ is to ensure the value-functions are well-defined, which is the case in fixed-horizon MDP. In general, I think it is okay to include it but I believe it adds unnecessary complication for reading.
> >
> > (4) Thank you for clarifying, so my understanding is that $\varepsilon$-equiprobability is more so addressing a numerical issue.
> >
> > (5) Yes, thank you for correcting me!
> >
> > I am happy to increase the score to weak accept after reading the rebuttal and other reviewers' comments. In particular I believe there can be a more difficult task that is not reaching. Perhaps a locomotion task with a specific velocity besides robotic manipulation: https://github.com/apexrl/GCRL-Collection

---

> > > ### Author Response · Authors · 2023-08-15
> > > **Thank you for increasing our score**
> > >
> > > Thank you for your response and increasing our score. We address the follow-up questions here:
> > >
> > > (1) The cell values are the reward function for the L2 shaping reward (Figure 2 (c)) but in the case of f-PG (Figure 2 (d)), these cannot be viewed as rewards because their values will change with the change in the state-visitation. It is true that the optimal behavior for the L2 shaped reward will be for the agent to stay at the non-negative state. But, since the states that take the agent outside the corner lead to a lower reward, the exploration will be hampered. But in the case of f-PG, the learning signals will weigh the actions taking the agent out of the corner more than the other actions.
> > >
> > > Weaknesses
> > >
> > > (1) Yes, we can use the policy gradient without $\gamma$ as well. Using $\gamma$ can help to stabilize training. We do want to point out that f-PG does not use value functions.

---

> > > > ### Comment · Reviewer_PJFh · 2023-08-15
> > > >
> > > > Thank you for the response, that clarifies the follow-up question.
> > > > Regarding $\gamma$, perhaps I overloaded the term and should have referred to the cumulative sum of rewards. That being said, thank you for indicating that $\gamma$ helps stabilize training.

---

> > > > > ### Author Response · Authors · 2023-08-20
> > > > >
> > > > > We appreciate the reviewer PJFh for their devoting time for the review and providing valuable suggestions. We hope that we have addressed all the concerns of the reviewer.

---

### Official Review · Reviewer_soXp · 2023-07-04

**Soundness:** 2 fair
**Presentation:** 2 fair
**Contribution:** 1 poor
**Rating:** 3
**Confidence:** 4

**Summary:**

The paper proposes to minimise the f-divergence $J(\theta) = D_f(p_\theta(s) \| p_g(s))$ between the desired state-distribution $p_g(s)$ and the current one $p_\theta(s)$. Clipped likelihood-ratio policy gradient is used to estimate the gradient. Kernel density estimation used to estimate the state visitations. Experiments on point reaching in grid world and in continuous-state maze are provided.

**Strengths:**

The paper is straightforward to follow. The idea and motivation are clear.
Originality: this niche has been explored in a number of works. The paper proposes a slightly novel approach, but its merit is not clear as it is only tested on simple point grid world and maze environments. In contrast, the baselines have been shown to work on robotics environments and scale quite well.
Quality: the paper is OK as a first step in validating the idea, but to really make claims about this algorithm, comparisons on harder environments are necessary. Important details are missing, e.g., it is not really clear from the paper how exactly the gradient is estimated and how KDE was used to estimate visitation frequencies, etc.
Clarity: the paper is mostly clear but it can be improved. E.g., the lemmas and theorems in Sec. 4 are obvious. It would be better to show Algorithm box there and provide more experiments.
Significance: currently quite low. Basically, experiments use f-KL, which is the well-known MaxEnt RL. So, the benefit of introducing the whole f-divergence story is unclear.

**Weaknesses:**

It is unclear how the authors deal with having $p_g(s)$ in the denominator of the ratio, since by definition it is a delta function. Some epsilon-modification is proposed for grid world, but it doesn't seem like it would scale to any reasonably high-dimensional task.

**Questions:**

See comments above

**Limitations:**

The paper does not provide a limitations section. Only future work is mentioned in the conclusion, but this is different from limitations. The authors are encouraged to add the limitations section.

---

> ### Author Rebuttal · Authors · 2023-08-09
>
> We thank the reviewers for lending their expertise to reviewing our paper, as well as for their constructive critique and suggestions for improvement.
> We have addressed several of the concerns in the main rebuttal as well.
>
> Strengths:
>
> (1) Comparison to harder environments: We have provided additional experiments in the rebuttal. We have discussed why we have restricted ourselves to simple domains in the main rebuttal (point 6).
>
> (2) We have explained the computation of KDE in the main rebuttal (point 5). We shall include an algorithm box in the camera ready version.
>
> (3) Some Lemmas in Section 4 might seem obvious but for the sake of completeness, we felt it pertinent to include them as well.
>
> (4) f-KL is not the same as MaxEnt RL. MaxEnt RL fails with sparse rewards even in the simplest of gridworlds (Section 5.1). f-KL maximizes the entropy of state-visitation while MaxEnt RL maximizes the entropy of policy which are different. We have discussed this further in the main rebuttal (point 3).
>
> Weakness:
>
> (1) The approximation should work in most cases. It is the same mathematical approximation to prevent divide by 0 that is commonly applied in many domains i.e. cut off the low values at an epsilon. We have discussed this approximation further in the main rebuttal.
>
> Limitations:
>
> Section 6 provides a combined discussion of the limitations and future work indicated by this approach. Taking the reviewer’s suggestion into account, we will flesh this section out in the camera ready version.

---

> > ### Comment · Reviewer_soXp · 2023-08-10
> > **Rebuttal Acknowledged**
> >
> > Thank you for your response and clarification, especially with regards to the relation to MaxEnt RL. It would be important in the future submission to explicitly contrast this method to MaxEntRL and to f-max / FAIRL in the paper since these are the two that immediately come to mind. And although I understand your point about the algorithm being on-policy, I believe it may be hard to publish unless you show that some version of your algorithm is competitive with the baselines on Walker or Hopper or more Fetch environments.

---

> > > ### Author Response · Authors · 2023-08-10
> > >
> > > Thank you for your response.
> > >
> > > (1) We have compared our results with f-AIRL in Section 5.1 and Section 5.2. For the gridworld in Section 5.1, the corresponding results for f-AIRL is in the supplementary. Also, the baselines in Section 5.1 is built on top of soft-Q Learning which follows from MaxEnt RL so we have comparisons with MaxEnt RL as well.
> > >
> > > (2) Walker and Hopper are not goal-conditioned environments. The other Fetch Environments have been used by recent works [1, 2] on top of an off-policy algorithm like DDPG or TD3 combined with HER. This shows that these environments are very difficult with on-policy methods.
> > >
> > > [1] I. Durugkar, et. al., Adversarial Intrinsic Motivation for Reinforcement Learning, NeurIPS 2021
> > >
> > > [2] B. Liu, et. al., Metric Residual Networks for Sample Efficient Goal-Conditioned Reinforcement Learning,

---

### Official Review · Reviewer_cLon · 2023-07-05

**Soundness:** 3 good
**Presentation:** 3 good
**Contribution:** 2 fair
**Rating:** 5
**Confidence:** 4

**Summary:**

This paper proposes a novel framework for goal-conditioned RL called f-Policy Gradients (f-PG), which minimizes the f-divergence between the agent's state visitation and the goal distribution to provide dense learning signals for exploration in sparse reward settings. The paper derives gradients for various f-divergences to optimize this objective and shows that entropy maximizing policy optimization for commonly used metric-based shaping rewards can be reduced to special cases of f-divergences. The paper compares f-PG with standard policy gradient methods on grid world environment as well as the Point Maze environments. It shows that f-PG performs well in all three environments and better than the baseline shaping rewards. The paper also illustrates how the learning signal evolves with the policy and performs an ablation to compare different f-divergences on their performances on the three Point Maze environments.

**Strengths:**

Firstly, a new framework for goal-conditioned RL which provides dense learning signals for exploration in sparse reward setting is proposed. Secondly, it is proved that minimizing the f -divergence (for some divergences) recovers the optimal policy. Furthermore, the analytical gradient for the objective looks very similar to a policy gradient which allow authors to use established methods from the policy gradient literature to come up with an efficient algorithm for goal-conditioned RL. Finally, empirical evidence in experiment part demonstrate that f-Policy Gradients outperform standard policy gradient methods and baseline shaping rewards in challenging environments. Overall, the paper presents a promising approach for addressing the exploration challenge in goal-conditioned RL.

**Weaknesses:**

In this paper, utilizing f-divergence to minimize the mismatch of an agent’s goal-conditioned state visitation to this target distribution is the main contribution. Besides, theoretical as well as empirical analysis has indicated the effectiveness of the methodology. However, some weaknesses still exist and remain to be polish up.

(1) Firstly, f-divergence is not a complete metric in math. Some shortcomings or limitations of it are not been discussed.

(2) Secondly, some notations in the paper are not explained clearly ( e.g. “trajectory dependent state visitation” in section 3, ).

(3) Furthermore, Experiment environments which are adopted in the paper seem to be too simple. Effectiveness of the methodology may not illuminated. Furthermore, some well-known methods in sparse reward setting are not compared.

(4) Novelty of the paper seems to be relatively weak.

While the main body of the paper is well-written, there is space for improvement. I defer some of my issues in the appendix to "Questions".


**Questions:**

Q1: What will be like if replacing f-divergence with some other mathematical complete metrics (i.e. L2-norm or wasserstein distance)? It is suggested to explain the motivation of utilizing f-divergence more.

Q2: How does f-PG compare to some famous baselines such as curiosity-driven[1] method and DIAYN[2]? It is suggested to add some Comparative experiment。

Q3: Ablation studies about different components of the methodology are suggested to added.

Q4: When the state of environment is in the form of image, is there any difference about the computation of f-divergence?

Q5: Some notations in the paper are not explained clearly ( e.g. “trajectory dependent state visitation” in section 3, ).

[1] Burda, Y., Edwards, H., Pathak, D., Storkey, A., Darrell, T., & Efros, A. A. (2018). Large-scale study of curiosity-driven learning. arXiv preprint arXiv:1808.04355.

[2] Eysenbach, B., Gupta, A., Ibarz, J., & Levine, S. (2018). Diversity is all you need: Learning skills without a reward function. arXiv preprint arXiv:1802.06070.



**Limitations:**

It would be better if the paper conducts more experiments from the aspects of ablation and comparison with other baselines. Whether effective or not in MARL setting may also be a good extension for the paper.

---

> ### Author Rebuttal · Authors · 2023-08-09
>
> We thank the reviewers for lending their expertise to reviewing our paper, as well as for their constructive critique and suggestions for improvement.
> We address each of the concerns raised by reviewer as follows:
>
> Weakness:
>
> (1) F-divergence is not symmetric but we are not sure how that will affect our learning process. In fact, value functions themselves are quasi-metrics [1].
>
> (2) Trajectory dependent state visitation is the frequency of states visited for the particular trajectory. Say for a 5 step trajectory, the states visited were s0, s1, s2, s2, s1. The state-visitation would be s0-> 1, s1-> 2 and s2->2.
>
> (3) We have discussed the issue of simple experiments in the section above on common questions (point 6). We have provided some additional experiments as well. We have compared our work with recent works that use a learnt shaping reward or use a distribution matching approach in reinforcement learning and imitation learning.
>
> (4) Our novelty is providing an alternative perspective to goal-conditioned reinforcement learning in terms of distribution matching. We do not need to learn a dense signal for augmenting the task reward rather the dense signals are provided by the f-divergence objective. We provide comprehensive proofs on why this objective makes sense. To the best of our knowledge, we are the first to propose this novel view for goal conditioned reinforcement learning.
>
> Questions
>
> (1) AIM uses Wasserstein distance. The benefit of f-divergence is inspired by the fact that minimizing f-divergence between distributions without overlap will push the distributions to have an overlap which means pushing for exploration. We have discussed the use of L2 in our common rebuttal.
>
> (2) Curiosity-driven methods mentioned in the cited paper use an encoder network to embed image observations and a dynamics model. We have not used visual inputs or model-based learning in our work so the comparison with curiosity-driven methods is inapplicable. DIAYN is an unsupervised reinforcement learning approach that discovers skills in an environment. It does not find the optimal policy for a particular task or goal so comparison with DIAYN is inappropriate.
>
> (3) We have compared f-PG for different f-divergences quantitatively in Section 5.4 and qualitatively in the supplementary material. We do not believe there are any other major ablations to perform on our method.
>
> (4) We have not dealt with vision based observations but using images as states can lead to several changes in the algorithm based on the environment. If the image states are assumed to be Markov, the KDE can be computed on encoded observations or use generative models to predict the probability of a sample in the distribution. Further modifications might be required, but it would be hard to predict these without empirical evaluation.
>
> (5) We explained the meaning of “Trajectory dependent state visitation” (point 2 of weaknesses).
>
> Limitations
>
> We have provided ablations and comparisons with the relevant baselines to the best of our knowledge. We have restricted our scope to single agent environments but it will be interesting to see if our method can be extended to a MARL setting.
>
> [1]: T Wang, et. al., Optimal Goal-Reaching Reinforcement Learning via Quasimetric Learning, ICML 2023

---

> > ### Comment · Reviewer_cLon · 2023-08-17
> >
> > First of all, thank you to the author for response to the questions which I raised. Some concerns have been answered adequately. However, the novelty of the paper is still relately weak, since  combining f-divergence with RL has been studied widely in RL domain. According to the author's response, I agree to increase the score of this article by 1 point.

---

> > > ### Author Response · Authors · 2023-08-17
> > > **Thank you Reviewer for increasing our score**
> > >
> > > We thank the reviewer for increasing our score.
> > >
> > > We would like to address the concerns over the novelty of our work.
> > >
> > > While there have been works that use f-divergences in imitation learning, the use of f-divergence in reinforcement learning is limited as discussed in our related work. The previous works that use f-divergence in imitation learning (against which we have compared our work), use the variational bound on f-divergence to reduce the training objective to a min-max objective which is solved by repeatedly switching between optimizing a discriminator and using this discriminator as the reward to optimize the policy by traditional RL algorithms (that assume the reward to be stationary) leading to unstable training (empirically verified as these methods have high-variance in their results).
> > >
> > > Our novelty lies in the following:
> > >
> > > (1) We show that minimizing the f-divergence between the agent's visitation distribution and the goal distribution yields the optimal policy.
> > >
> > > (2) While previous methods still rely on the traditional RL algorithms that assume access to stationary rewards to optimize the policy, we directly optimize the policy to minimize the f-divergence without using rewards. The gradient of the f-divergence looks similar to a policy gradient weighted by learning signals from the f-divergence which are neither rewards nor are they assumed to be stationary.
> > >
> > > (3) These learning signals implicitly push the agent to explore until a significant coverage between the agent's state visitation distribution and the goal distribution is achieved. The discriminator-based baselines fail when there isn't much coverage between the agent's state visitation distribution and the goal distribution.
> > >
> > > (4) f-PG yields a mathematically stable algorithm which can be empirically observed from the low variance in our results compared to the discriminator-based baselines.

---

> > > > ### Author Response · Authors · 2023-08-20
> > > >
> > > > We appreciate the reviewer cLon for their devoting time for the review and providing valuable suggestions. We hope that we have addressed all the concerns of the reviewer. We would like to reiterate our response to the concerns over novelty,
> > > >
> > > > (1) Previous methods have focussed on using f-divergence for imitation learning. Few methods like Dual-DICE[1], have used f-divergence as a regularizer with an already available reward function rather than as a learning signal.
> > > >
> > > > (2) We prove that minimizing the f-divergence between the agent’s state visitation and the goal distribution yields the optimal policy.
> > > >
> > > > (3) We provide a mathematically stable objective which, unlike previous methods, do not assume stationary rewards while policy optimization.
> > > >
> > > > (4) The learning signals incentivizes exploration and does not assume coverage between the goal distribution and the agent’s visitation without which the discriminative methods fail.
> > > >
> > > > [1] O. Nachum, et. al., DualDICE: Behavior-Agnostic Estimation of Discounted Stationary Distribution Corrections

---

### Author Rebuttal · Authors · 2023-08-09

We thank the reviewers for their reviews. We observe that the reviewers have unanimously agreed to the novelty of the f-divergence perspective of goal conditioned RL. They agree that this method is a promising approach for solving the exploration challenge in sparse reward RL.
We respectfully submit that, compared to reviews on other papers we have seen both this year and in past years,  the numerical scores given by the reviewers are significantly lower than is indicated by the text of the reviews. As detailed below, most of the questions and comments were clarification questions that we can easily address, or are requests to compare with baselines that are not directly comparable to our work.

Please recall that the main contribution of our paper is providing a new perspective on goal conditioned RL by posing it as a distribution matching problem. To our knowledge, we are the first to provide a formal proof that minimizing the f-divergence between the agent’s visitation distribution and the goal distribution produces the optimal policy. The divergence minimization objective provides dense signals for policy optimization, unlike the traditional reward maximization perspective which struggles to learn in sparse reward settings.

We now reiterate a short comparison with related work to emphasize why this approach is novel and valuable. Some prior works have looked at imitation learning as a distribution matching problem but model the problem using minmax objectives which necessarily require discriminators. The commonly used MaxEnt RL can be thought of as minimizing the KL divergence between the agent’s trajectory distribution and a hypothetical optimal trajectory distribution represented using the exponent of the return. MaxEnt RL can be shown to maximize the reward as well as policy entropy, a tradeoff that is difficult to tune in sparse reward settings. Several methods exist that learn an additional reward (referred to as shaping reward) that can be augmented to the task reward. Our work differs from MaxEnt RL as we do not hypothesize a target trajectory distribution – rather we assume a goal distribution and we minimize the f-divergence between the agent’s state visitation distribution and the goal distribution. We also do not learn an additional reward signal to optimize the policy, rather use the dense learning signals from the f-divergence to directly optimize the policy.

There were several common clarification questions which we answer here.

(1) How did we use the Dirac distribution for the goal distribution as the f-divergence will not make sense?
In the proofs of Lemma 4.2 and Theorem 4.1, we assume the goal distribution to be a dirac distribution pointing out that this might not be true for all f-divergences. In fact, f-divergences can be defined between two distributions that do not overlap if $f’(\infty)$ is defined. For the computation of the f-policy gradient, we show in Theorem 4.3, that a dirac distribution at the goal will lead to the true policy gradient while a more diffuse target distribution will lead to the gradient in Theorem 4.2. We make an approximation for the Dirac distribution by cutting off the zero probability at \epsilon only in our experiments, so that we can use the gradient from Theorem 4.2. This is a standard approximation used to prevent divide by zero instabilities in several implementations. Our approximation is further explained in Figure 1 of the rebuttal.

(2) Does using an $\epsilon$ approximation for the Dirac distribution make it same as uniform reward shaping?
No. A uniform reward is not the same as the $\epsilon$ approximation. The details are further explained in Figure 1 of the rebuttal. A uniform reward will not push the agent to explore unvisited states while f-PG pushes the agent to increase the entropy of the state-visitation distribution. It will thus act as an impetus for the agent to explore more effectively.

(3) Is FKL the same as MaxEnt RL?
FKL is not the same as MaxEnt RL. MaxEnt RL aims to maximize the entropy of the policy while FKL will increase the entropy of the agent’s state-visitation distribution. These are not the same as shown in Figure 2.

(4) How does L2 shaping reward compare with f-PG?
L2 shaping reward incentivises the agent to move along the line segment connecting the agent’s state and the goal state. In maze situations, L2 shaping reward fails terribly as the optimal path often does not involve following the line segment from the current state to the goal state. We have explained this further in Figure 2.

(5) How is the agent’s state-visitation distribution computed?
As mentioned in Section 4.3, we use kernel density estimation to compute the agent’s state visitation. For a given goal distribution, we collect several trajectories $\tau_i ; g$. We fit a kernel density estimator on the states in these trajectories $s \in {\tau_i ; g}$. This gives $p_\theta(s)$ for the goal distribution $p_g(s)$ which is computed by fitting a kernel density estimator on the samples from $g$.

(6) Why do the experiments look simple?
One limitation of our algorithm (as discussed in the paper) is that it is on-policy. To the best of our knowledge, all the traditional robotics benchmarks like Fetch or ShadowHand use an off-policy algorithm like DDPG or TD3. Off-policy algorithms like these are considerably more sample efficient and can make use of samples from exploratory policies, while an on-policy algorithm can only use samples from policies that are very close to the current policy. We have provided some additional experiments on a complex PointMaze and FetchReach in the rebuttal in Figure 3. Extending this approach to more difficult domains is an exciting direction for future research. However, it will involve augmenting f-PG to allow for off-policy updates, which is out of the scope of this paper.

---

### Decision · Program_Chairs · 2023-09-21

**Decision:**

Accept (poster)

**Comment:**

This paper proposes f-divergence as a new theoretical framework for goal-conditioned RL problems. The main theoretical result is that minimising f-divergence between the agent's state visitation distribution and the goal yields an optimal policy. The gradient of this f-divergence naturally gives a dense learning signal for the policy. The results on PointMaze and FetchReach (added during the rebuttal period) show competitive performance to some of the existing methods such as AIM, GAIL, and F-AIRL.

Most of the reviewers acknowledged that the theoretical result is quite novel and interesting. In the meantime, some of the reviewers remained not convinced by the practicality of the proposed method as it is on-policy, which tends to be fundamentally less data-efficient than off-policy methods, as acknowledged by the authors. Nevertheless, I found that the new theoretical framework and its connection to max-entropy RL with reward shaping based on L2 distance are promising and novel enough to be presented at the NeurIPS community. Thus, I recommend accepting this paper. I also suggest the authors point out the distinction from the previous work on f-divergence in RL such as [1] in the revision.

[1] Ke et al., Imitation Learning as f-Divergence Minimization